# Microbiome homeostasis on rice leaves is regulated by a precursor molecule of lignin biosynthesis

Pin Su [1,10], Houxiang Kang [2,10], Qianze Peng [3,4,10], Wisnu Adi Wicaksono [5], Gabriele Berg [5,6,7], Zhuoxin Liu[8], Jiejia Ma[8], Deyong Zhang [1,3,4] ✉, Tomislav Cernava [5,9] ✉ & Yong Liu[1] ✉

In terrestrial ecosystems, plant leaves provide the largest biological habitat for highly diverse microbial communities, known as the phyllosphere microbiota. However, the underlying mechanisms of host-driven assembly of these ubiquitous communities remain largely elusive. Here, we conduct a large-scale and in-depth assessment of the rice phyllosphere microbiome aimed at identifying specific host-microbe links. A genome-wide association study reveals a strong association between the plant genotype and members of four bacterial orders, Pseudomonadales, Burkholderiales, Enterobacterales and Xanthomonadales. Some of the associations are specific to a distinct host genomic locus, pathway or even gene. The compound 4-hydroxycinnamic acid (4-HCA) is identified as the main driver for enrichment of bacteria belonging to Pseudomonadales. 4-HCA can be synthesized by the host plant's OsPAL02 from the phenylpropanoid biosynthesis pathway. A knockout mutant of *OsPAL02* results in reduced Pseudomonadales abundance, dysbiosis of the phyllosphere microbiota and consequently higher susceptibility of rice plants to disease. Our study provides a direct link between a specific plant metabolite and rice phyllosphere homeostasis opening possibilities for new breeding strategies.

In nature, plants and their associated microbes, collectively known as the microbiota, form functional entities that rely on each other[1]. The microbiota contributes to aspects such as disease resistance[2–5], stress tolerance[6,7], and nutrient acquisition[8,9]. Successful recruitment and maintenance of a sufficient abundance of specific microbial members determines the outcome of plant-microbiota interactions[10,11]. Thus,

understanding the principles driving microbiome assembly in crop plants has become one of the main pursuits of present studies in order to integrate microbiome functioning into sustainable crop production.

The plant microbiota can be compartmentalized into the rhizosphere (root-soil interface), phyllosphere (leaf surface) and endosphere (internal tissues)[12]. Studies revealed that host genetics[13] and metabolic

[1]State Key Laboratory of Hybrid Rice and Institute of Plant Protection, Hunan Academy of Agricultural Sciences, Changsha 410125, China. [2]State Key Laboratory for Biology of Plant Diseases and Insect Pests, Institute of Plant Protection, Chinese Academy of Agricultural Sciences, Beijing 100193, China. [3]National Center of Technology Innovation for Saline-Alkali Tolerant Rice in Sanya City, Sanya 572024, China. [4]College of Tropical Crops, Hainan University, Haikou 570228, China. [5]Institute of Environmental Biotechnology, Graz University of Technology, Graz 8010, Austria. [6]Leibniz Institute for Agricultural Engineering and Bioeconomy (ATB), Potsdam 14469, Germany. [7]Institute for Biochemistry and Biology, University of Potsdam, Potsdam 14476, Germany. [8]Longping Branch, College of Biology, Hunan University, Changsha 410082, China. [9]School of Biological Sciences, Faculty of Environmental and Life Sciences, University of Southampton, Southampton SO17 1BJ, UK. [10]These authors contributed equally: Pin Su, Houxiang Kang, Qianze Peng. ✉e-mail: zhangdeyong@hhrrc.ac.cn; tomislav.cernava@tugraz.at; liuyong@hunaas.cn

processes[14] participate in the recruitment of specific microbial taxa in the rhizosphere microbiome, and consequentially shape microbiome assembly. In barley, a specific genomic locus was shown to be associated with the recruitment of specific bacterial taxa[15]. Deepening analysis revealed three genes in this locus, including a Nucleotide-Binding-Leucine-Rich-Repeat (NLR) receptor as primary candidates involved in shaping the microbiome[15]. In tomato plants, the identification of specific quantitative trait loci (QTLs) provided further evidence for the importance of host genetics in microbiome assembly[16]. In addition, specific plant root exudates were demonstrated to steer rhizosphere microbiome assembly toward the host's needs for defense and nutrition[17]. The plant phyllosphere represents the largest biological surface on earth and widespread interface for interactions between plants and their microbiota[18]. Identification of specific genetic components shaping the phyllosphere microbiome is still sparsely understood, although the plant genotype has been repeatedly demonstrated to impact the phyllosphere microbiome in consistent ways across geographically separated sites[19,20]. Model plants like *Arabidopsis*[21] and non-model plants like perennial wild mustard[22] as well as cereal crops such as wheat and barley[23] were implemented to increase our understanding of the phyllosphere microbiome.

It is known that external factors such as microbial inocula and climate conditions can alter phyllosphere microbiomes[24], however, the plant genotype has become the central focus in terms of seeking approaches to engineer plant microbiome for agricultural purposes[25]. Recent studies have revealed that defense responses and cell wall integrity are likely involved in phyllosphere microbiome assembly[19]. *Arabidopsis* cuticle mutants and ethylene signaling mutants were shown to have a different phyllosphere microbiome compared to wild-type plants[20]. The phosphate starvation response pathway[26] and pattern-triggered immunity[27] were also shown to take part in shaping the *Arabidopsis* phyllosphere microbiome. A common strategy to investigate host genetic effects on microbiome assembly is based on amplicon sequencing of microbial marker genes coupled with genome-wide association studies (GWAS)[28]. Such studies have resulted in the identification of many putatively causal genes with effects on the relative abundance of plant-associated microbes[19,21,22]. In the case of the *Arabidopsis* phyllosphere microbiome, specific microbial hubs were successfully linked to host genomic loci that are related to specialized metabolite biosynthesis[21]. Genes involved in the synthesis of sinapoyl glucose, sinapoyl malate, glucosinolates are assumed to participate in shaping the *Arabidopsis* phyllosphere microbiome. Nevertheless, there is still lack of direct evidence bridging the host genetic background with the recruitment of specific microbial members. So far, phenotyping phyllosphere microbiome compositions via GWAS was exclusively conducted on the basis of amplicon sequencing of microbial marker genes[17]. However, metagenomic data provides better means to more accurately conduct such analyses, mainly because it is less prone to over- or underestimate the abundance of certain microbial taxa.

In this work, with the employment of phyllosphere metagenomes from 110 rice accessions of the Rice Diversity Panel II core collection (C-RDP-II)[29], we perform GWAS experiments to link bacterial abundances with single nucleotide polymorphisms (SNPs) in the rice genome. Rice genetic variations are shown to be significantly associated with members of four predominant phyllosphere bacterial orders, Pseudomonadales, Burkholderiales, Enterobacterales, and Xanthomonadales. To unravel a prevailing mechanism of how host genetics affect phyllosphere microbiome assembly, we implement mutants and over-expression constructs of a candidate gene associated with Pseudomonadales and assess the resulting microbiome shifts. Furthermore, we analyze rice metabolites in rice leaves that are regulated by the candidate gene. We discover that the compound 4-hydroxycinnamic acid, also known as p-coumaric acid and a precursor in lignin biosynthesis, is required for the assembly and homeostasis of the rice phyllosphere microbiome. Overall, our study provides clear evidence for host metabolite-driven phyllosphere microbiome assembly.

## Results

### Rice phyllosphere microbiomes are specific to genotypes

A large-scale dataset was implemented to assess microbiome structures in the phyllosphere of rice plants. The dataset was obtained via shotgun-based metagenomics[29]. It is based on 110 accessions of the Rice Diversity Panel II core collection (C-RDP-II) that were all grown in the northwest of Hunan Province, China (28°38′09″ N, 112°0′57″ E). For comparative analyses, the rice accessions were separated into three main groups: 56 *indica*, 36 *japonica* and 18 unassigned (Supplementary Data 1). Rarefaction analysis confirmed that the implemented approach sufficiently captured microbial diversity in the *indica* and *japonica* groups (Supplementary Fig. 1). Processing of the metagenomes using Kraken2 and Bracken[30] generated 6,862 species-level taxonomic units. Bacteria (comprising over 94% of the total taxonomic units) were the prevalent microbiome constituents in the rice phyllosphere (Supplementary Data 2).

We found that bacterial communities were specific to rice subspecies. Bacterial communities obtained from *indica* and *japonica* varieties formed two distinct clusters, as indicated by the unconstrained principal coordinate analysis (PCoA) of Bray-Curtis distances (Fig. 1a, $P < 0.001$, $R^2 = 0.03$, PERMANOVA with Adonis test). This suggests that the genotypes of *indica* and *japonica* varieties are connected to distinct phyllosphere microbiome compositions and structures. Furthermore, we assessed the within-sample diversity (α-diversity) and found a significant difference between *indica* and *japonica* varieties (Fig. 1b and Supplementary Fig. 2). The phyllosphere microbiome of *indica* was more diverse than that of *japonica* (Fig. 1b and Supplementary Fig. 2), indicating that the former was colonized by more bacterial species. We also observed significant differences in the phyllosphere microbiome structures between *indica* and *japonica* varieties at the phylum, order and genus levels (Supplementary Fig. 3, 4, Fig. 1c, Supplementary Table 1), providing further evidence that bacteria are influenced by the rice genotypes.

The predominant bacterial orders of both *indica* and *japonica* rice plants were comprised of Pseudomonadales (57% and 72% of relative abundance for *indica* and *japonica*, respectively), followed by Enterobacterales (19% and 7%), Sphingomonadales (5% and 5%), Burkholderiales (3.7% and 2.1%), Xanthomonadales (1.8% and 3.5%), Flavobacteriales (3.4% and 1.1%), Rhizobiales (1.5% and 1.3%), Caulobacterales (1.3% and 1.7%) and Bacillales (1.7% and 0.8%) (Fig. 1c). Compared to *japonica*, *indica* varieties had a higher relative abundance of Enterobacterales, Flavobacteriales and Burkholderiales, while *japonica* varieties had a higher relative abundance of Pseudomonadales and Xanthomonadales (Tukey's HSD test, false discovery rate (FDR) adjusted $P < 0.05$) (Fig. 1c, Supplementary Table 2). Interestingly, 46 genera from Enterobacterales (accounting for 92% of the identified genera within this order), 17 genera from Burkholderiales (27.4% of the identified genera) and 5 genera from Pseudomonadales (83% of the identified genera) occurred at significantly different relative abundances in *indica* and *japonica* varieties (Supplementary Fig. 4, Supplementary Data 2). In contrast, only the genus *Aerosticca* from Xanthomonadales and only the genus *Chryseobacterium* from Flavobacteriales showed such differences in relative abundance between the varieties. Therefore, we hypothesized that a specific genetic background of the rice varieties may play a role in shaping their phyllosphere microbiomes.

### Identification of metabolic pathways linked to phyllosphere microbiome

With recent advances in sequencing and bioinformatics, genome-wide association studies (GWAS) have become an increasingly promising

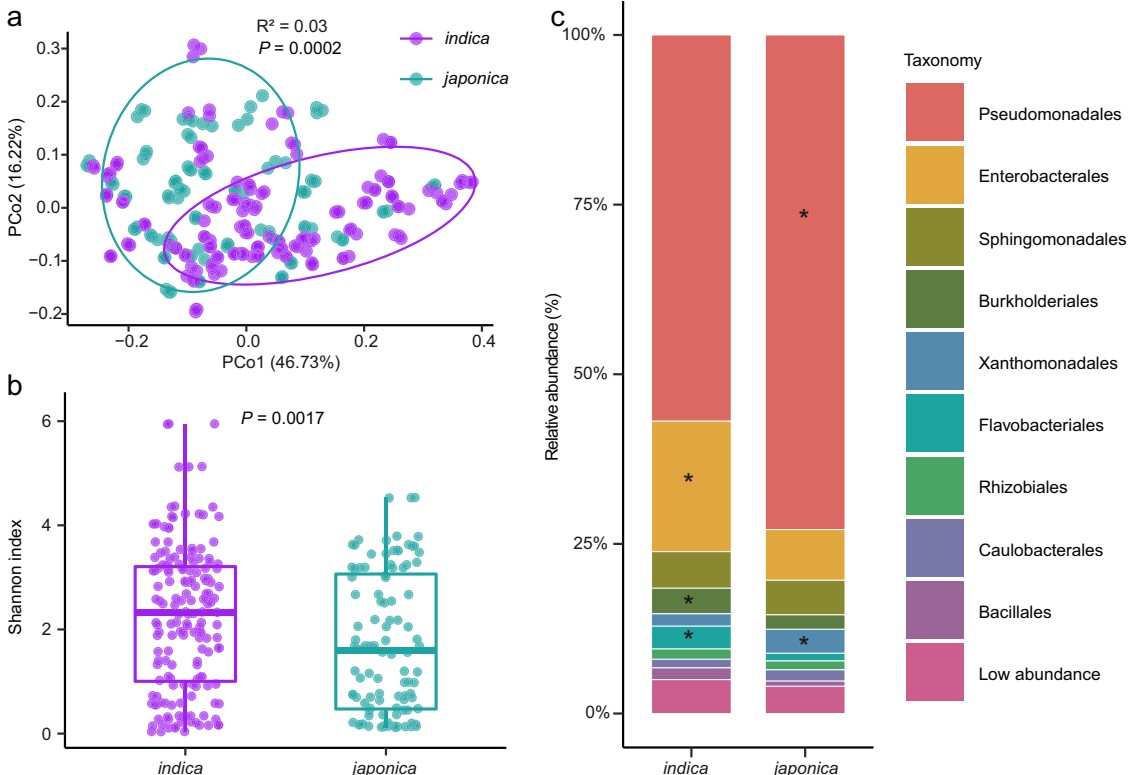

**Fig. 1 | Phyllosphere microbiome comparisons of *indica* and *japonica* rice varieties. a** Unconstrained PCoA (for principal coordinates PCo1 and PCo2) based on Bray-Curtis distances showing bacterial community clustering of *japonica* and *indica* rice varieties ($P < 0.001$, $R^2 = 0.03$, *P*-value was calculated by one-way PER-MANOVA). Ellipses cover 68% of the data for each rice subspecies. **b** Shannon index for phyllosphere bacterial communities of *indica* and *japonica* varieties. The horizontal bars within boxes represent medians. The tops and bottoms of boxes represent the 75th and 25th percentiles, respectively. The upper and lower whiskers extend to data no more than 1.5× the interquartile range from the upper edge and lower edge of the box, respectively. The *P*-value was calculated with unpaired one-way ANOVA with Tukey's HSD test ($P < 0.05$). **c** Order-level distribution of bacteria in the *indica* and *japonica* phyllosphere microbiomes. Asterisks represent significant differences between *indica* and *japonica* varieties as assessed with unpaired one-way analysis of variance (ANOVA) with Tukey's HSD test ($P < 0.05$, *P*-values are listed in Supplementary Table 2). *indica* ($n = 56$) and *japonica* ($n = 36$) with three replications for each genotype. Source data are provided as a Source Data file.

approach for identifying genetic factors associated with important agronomic traits in plants[28]. To identify potential rice genetic factors that are linked to the phyllosphere microbiome, we conducted GWAS. The analysis targeted associations between rice genomes and the relative abundance of bacterial species within the respective metagenomes. Our analysis led to the identification of 2,667 non-redundant single nucleotide polymorphisms (SNPs) located in 235 loci that were significantly associated with 496 bacterial species (hereafter referred to as GWAS-associated species) at a genome-wide significance threshold of $P \le 0.00001$ (Fig. 2c, Supplementary Data 3).

We found that the majority of GWAS identified species belonged to Pseudomonadales (associated with 129 loci, accounting for 30.24% of total GWAS-associated species) Burkholderiales (associated with 100 loci, accounting for 21.37%), Xanthomonadales (associated with 47 loci, accounting for 8.87%) and Enterobacterales (associated with 76 loci, accounting for 8.67%) at the bacterial order level (Fig. 2a, Supplementary Data 4). To separate loci according to bacterial orders associated with them, we searched for locus clusters using Bray-Curtis distances based on GWAS-associated species distribution of each locus. Our results showed that 91.06% of the total SNP loci could be divided into four clusters: Cluster1 (containing 38.30% of total SNP loci and associated with Pseudomonadales), Cluster2 (containing 31.91% of total SNP loci and associated with Burkholderiales), Cluster3 (containing 12.77% of total SNP loci and associated with Enterobacterales) and Cluster4 (containing 8.09% of total SNP loci and associated with Xanthomonadales) (Fig. 2a, Supplementary Fig. 5, Supplementary Data 4). These associations suggested that members of four bacterial

orders are particularly responsive to certain genetic backgrounds of the host plant.

To further elucidate how host genetic variations affect the phyllosphere microbiome, especially members of the bacterial orders that were highly responsive, genes with SNPs located in the associated loci were annotated and enriched into rice pathways (Supplementary Data 4). We found that metabolic pathways (dosa01100) and secondary metabolic process (GO:0019748) were prevalent in gene-enriched KEGG (Kyoto Encyclopedia of Genes and Genomes) (Supplementary Fig. 6) and GO (Gene Ontology) (Supplementary Fig. 6) annotations, respectively. Furthermore, we observed that SNP-associated genes in each locus cluster were enriched in specific metabolic pathways (Fig. 2b). For instance, Enterobacterales-associated rice genes were significantly enriched in Proteasome (dosa03050), Lysine degradation (dosa00310), Biosynthesis of amino acids (dosa01230), Ubiquitin mediated proteolysis (dosa04120), SNARE interactions in vesicular transport (dosa04130), Cysteine and methionine metabolism (dosa00270), Phosphatidylinositol signaling system (dosa04070), and Glutathione metabolism (dosa00480) pathways (Fisher's exact test, $P < 0.05$, Fig. 2b, Supplementary Data 4). Similarly, Burkholderiales-associated rice genes were significantly enriched in Pentose and glucuronate interconversions (dosa00040), Protein export (dosa03060), Phenylalanine metabolism (dosa00360) and Arginine and proline metabolism (dosa00330) pathways (Fisher's exact test, $P < 0.05$, Fig. 2b, Supplementary Data 4). Pseudomonadales-associated rice genes were significantly enriched in Plant-pathogen interaction (dosa04626), Stilbenoid, diarylheptanoid and gingerol biosynthesis

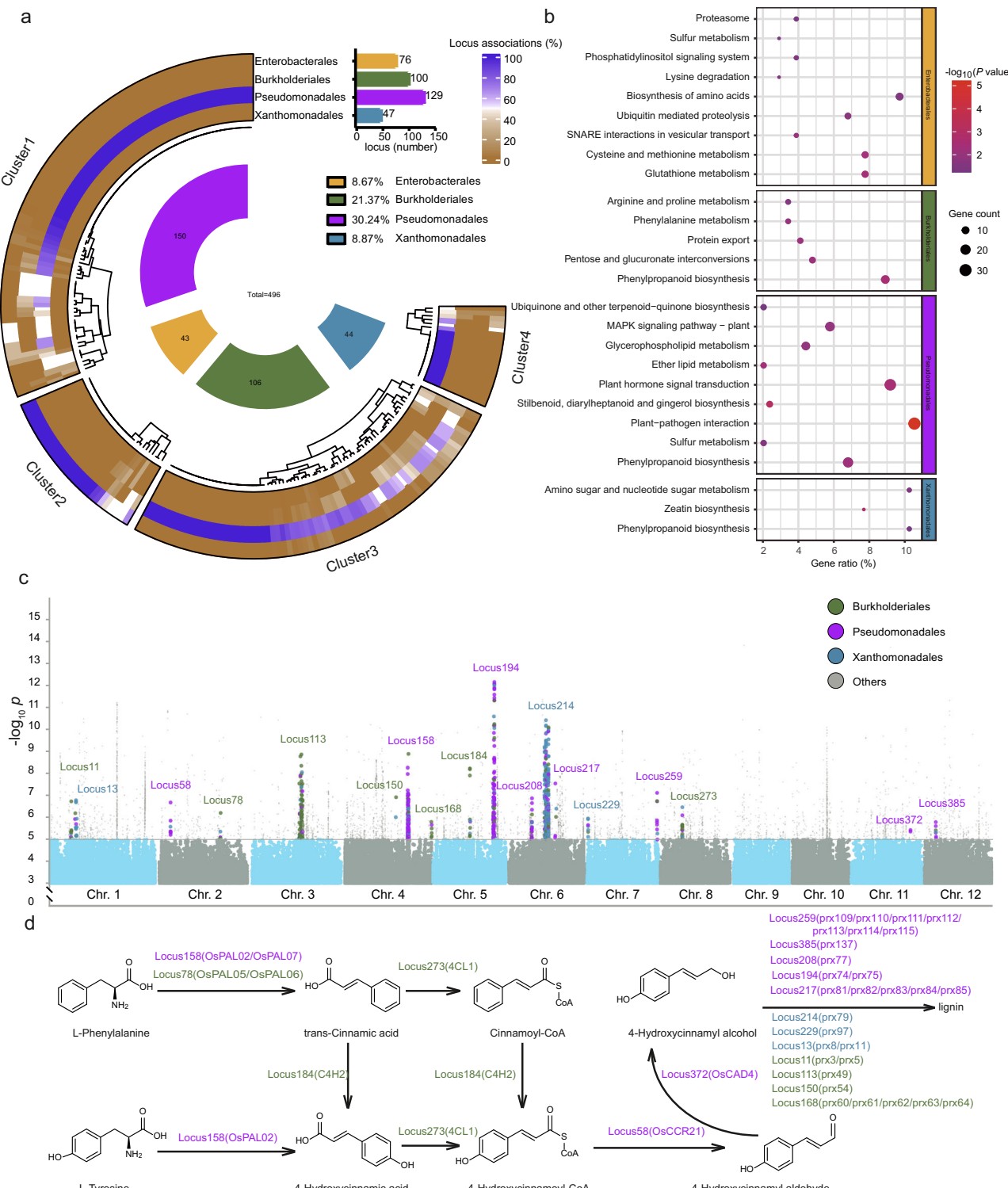

**Fig. 2 | GWAS with rice phyllosphere microbiome. a** Species associations for the four predominant bacterial orders and GWAS locus clusters. GWAS locus clustering is based on Bray-Curtis distances of bacterial species associated with each locus. Cluster1 (containing 38.30% of total SNPs; mainly associated with members of Pseudomonadales), Cluster2 (31.91%; mainly associated with members of Burkholderiales), Cluster3 (12.77%; mainly associated with members of Enterobacterales) and Cluster4 (8.09%; mainly associated with members of Xanthomonadales) (Supplementary Data 4). The total numbers of linked GWAS loci are shown for each order in the corresponding bar plot. The inner segments indicate the taxonomic group of the GWAS-associated bacterial species. Numbers in the inner segments indicate associated bacterial species and are color-coded according to their order-level annotation. **b** KEGG enrichment analysis for genes from locus clusters in **a**. The color and size of the bubbles represent the P-value (-log₁₀, two-sided Fisher's exact test, P < 0.05, P-values are listed in Supplementary Data 4) and number of enriched genes in each KEGG pathway, respectively. **c** Manhattan plots of GWAS results obtained with 110 rice varieties. GWAS loci and SNP plots are color-coded according to bacterial order-level annotation. Two-sided P-values were determined using canonical correlation analysis. **d** Analysis of genes that were enriched in the phenylpropanoid biosynthesis pathway. GWAS locus and gene names are color-coded according to bacterial order-level annotation. Source data are provided as a Source Data file.

(dosa00945), Plant hormone signal transduction (dosa04075), Ether lipid metabolism (dosa00565), Glycerophospholipid metabolism (dosa00564), MAPK signaling pathway-plant (dosa04016) and Ubiquinone and other terpenoid-quinone biosynthesis (dosa00130) pathways (Fisher's exact test, $P < 0.05$, Fig. 2b, Supplementary Data 4). Xanthomonadales-associated rice genes were significantly enriched in Zeatin biosynthesis (dosa00908) and Amino sugar and nucleotide sugar metabolism (dosa00520) (Fisher's exact test, $P < 0.05$, Fig. 2b, Supplementary Data 4). Notably, there were also overlapping pathways among different bacterial orders. For example, Enterobacterales- and Pseudomonadales-associated rice genes were enriched in the Sulfur metabolism pathway (dosa00920), while Burkholderiales-, Pseudomonadales- and Xanthomonadales-associated rice genes were enriched in the Phenylpropanoid biosynthesis pathway (dosa00940) (Fig. 2b, Supplementary Data 4). These results provided key insights into host metabolic pathways potentially involved in the assembly of the phyllosphere microbiota. Since the Phenylpropanoid biosynthesis pathway was associated with three predominant bacterial orders, we expect a key role of this pathway with respect to host-microbiota interactions.

We therefore further investigated Burkholderiales-, Pseudomonadales- and Xanthomonadales-associated rice genes that were enriched in the Phenylpropanoid biosynthesis pathway (Fig. 2c, d). We found that a majority of *prx* (peroxidase) genes, which are involved in lignin biosynthesis, were associated with Burkholderiales, Pseudomonadales and Xanthomonadales (Fig. 2d, Supplementary Data 4). In addition, the *4CL1* (4-coumarate-CoA ligase, Os08g0245200) and *C4H2* (cinnamate-4-hydroxylase, Os05g0320700) genes, which are involved in the synthesis of cinnamoyl-CoA and 4-hydroxycinnamoyl-CoA, respectively, were found to be specifically associated with Burkholderiales. On the other hand, *OsPAL02* (phenylalanine/tyrosine ammonia-lyase, Os04g0518100), *OsCAD4* (cinnamyl alcohol dehydrogenase, Os11g0622800) and *OsCCR21* (cinnamoyl-CoA reductase, Os02g0180700) genes, which are involved in the synthesis of 4-hydroxycinnamic acid, 4-hydroxycinnamyl alcohol and 4-hydroxycinnamyl aldehyde, respectively, were specifically associated with Pseudomonadales (Fig. 2d, Supplementary Data 4). This indicated potential differences in host genetic control of the phyllosphere microbiota. Collectively, our analysis suggested that compounds that are part of lignin biosynthesis or their precursors may be participating in regulating microbiota assembly, and more importantly, that certain compounds may exert effects on specific bacterial taxa.

## Identification of leaf metabolite-associated genes that drive microbiome assembly

Given the common occurrence and already known significance of members from the Pseudomonadales order in the plant phyllosphere, we took particular interest in deciphering how plant genetics may be involved in the enrichment of it. Pseudomonadales were not only highly abundant in the present study, but also accounted for the largest number of associated loci in the GWAS approach, indicating a robust connection to plant genetics. To verify rice genetic factors that affect the assembly of Pseudomonadales, we performed gene-editing to mutate a host gene shown to be specifically associated with it. *OsPAL02* was selected as the target gene due to the strong association Pseudomonadales (*P*-value: 1.33E-09) and because there are no paralogous genes in locus 158 and other loci identified by GWAS. Using 16S rRNA gene fragment amplicon sequencing, we showed that the rice *OsPAL02* gene mutant (hereafter referred to as *OsPAL02-KO*), which lacks phenylalanine/tyrosine ammonia-lyase, showed a significant change in the relative abundance of Pseudomonadales (Fig. 3a). Intriguingly, natural variation of SNPs in *OsPAL02* was different between *indica* and *japonica* rice (Supplementary Fig. 7), suggesting a correlation between Pseudomonadales abundance and this gene.

To further investigate the detailed role of *OsPAL02* in shaping the rice leaf microbiota, we carried out 16S rRNA gene fragment amplicon sequencing to profile the leaf microbiota of wild-type (WT) rice variety ZH11 (*japonica*), the loss-of function mutant, *OsPAL02-KO* and the over-expression construct, *OsPAL02-OE*. Under greenhouse conditions, we observed that the community structure in the phyllosphere of WT, *OsPAL02-OE* and *OsPAL02-KO* lines were dominated by Pseudomonadales, Enterobacterales, Sphingomonadales, Xanthomonadales, Flavobacteriales, Paenibacillales, Rhizobiales, Sphingobacteriales and Burkholderiales (Fig. 3a). This result was consistent with initial analysis of the leaf microbiome, indicating that field observations were reproducible in the greenhouse. Unconstrained PCoA (Fig. 3b, $P < 0.001$, $R^2 = 0.32$, PERMANOVA) and constrained principal coordinate analysis (CPCoA) (Supplementary Fig. 8, 23.7% of total variance was explained by the plant genotype, $P < 0.001$, PERMANOVA) of Bray-Curtis distances revealed that the WT, *OsPAL02-OE* and *OsPAL02-KO* phyllosphere microbiomes formed separate clusters, indicating that OsPAL02 affected their bacterial communities. Further analysis of differences in phyllosphere microbiomes between WT, *OsPAL02-OE* and *OsPAL02-KO* revealed significant variations at the order level (Fig. 3a). The relative abundance of Pseudomonadales showed a significant decrease in *OsPAL02-KO* and increase in *OsPAL02-OE*, compared to WT plants respectively (Fig. 3c). The relative abundance of Xanthomonadales, Flavobacteriales and Burkholderiales showed a significant increase in *OsPAL02-KO* compared to WT or *OsPAL02-OE* plants (Fig. 3d, Supplementary Table 3, Supplementary Data 5). Notably, the Shannon index (alpha-diversity) of *OsPAL02-KO* was significantly higher than that of *OsPAL02-OE* and WT (Supplementary Fig. 8). Similar observations were made when two additional *OsPAL02-KO* and *OsPAL02-OE* lines were analyzed (Supplementary Fig. 9, Supplementary Tables 4 and 5, Supplementary Data 5). The observation that *OsPAL02-KO* harbored higher alpha-diversity and lower relative abundance of Pseudomonadales than wild-type ZH11 (*japonica*) followed the same trend observed with *indica* and *japonica* varieties (Fig. 1b, c). This finding indicated that differences in *OsPAL02* were sufficient to mimic microbiome variations between *indica* and *japonica* plants. Conclusively, the gene *OsPAL02* was shown to contribute to the recruitment of Pseudomonadales and shape the bacterial community structure in the rice phyllosphere.

In the phenylpropanoid biosynthesis pathway, the gene *OsPAL02* encodes a phenylalanine/tyrosine ammonia-lyase which catalyzes the transformation of L-tyrosine into 4-hydroxycinnamic acid (4-HCA) as well as the transformation of L-phenylalanine into trans-cinnamic acid, which are both precursors for the biosynthesis of lignin. When *OsPAL02* variation between *japonica* and *indica* was analyzed, the protein structures of haplotype 1 (mainly present in *indica*) and haplotype 5 (mainly present in *japonica*) showed an amino acid exchange at site 134. This result indicated that OsPAL02 catalytic activity may differ between *japonica* and *indica* plants (Supplementary Fig. 7). We, therefore, hypothesized that the products of OsPAL02 impose a selective force on the recruitment of phyllosphere microbiota members, especially those within the bacterial order Pseudomonadales.

To investigate this hypothesis, we performed metabolite analyses with rice leaves of WT, *OsPAL02-OE*, and *OsPAL02-KO*, in order to identify differences attributable to *OsPAL02*. The results showed that WT, *OsPAL02-OE*, and *OsPAL02-KO* metabolite profiles were separated into three clusters based on Bray-Curtis distances and confirmed that *OsPAL02* was involved in the biosynthesis of cinnamic acid and its derivatives (Fig. 4a, Supplementary Data 6). The data was also subjected to unconstrained PCoA and the results showed that the first principal coordinates PCo1, which accounts for 53.89% of the total variance, separated the WT, *OsPAL02-OE*, and *OsPAL02-KO* plants, indicating substantial differences in metabolites present in rice leaves (Fig. 4b, $P < 0.001$, $R^2 = 0.60$, PERMANOVA). This was also observed

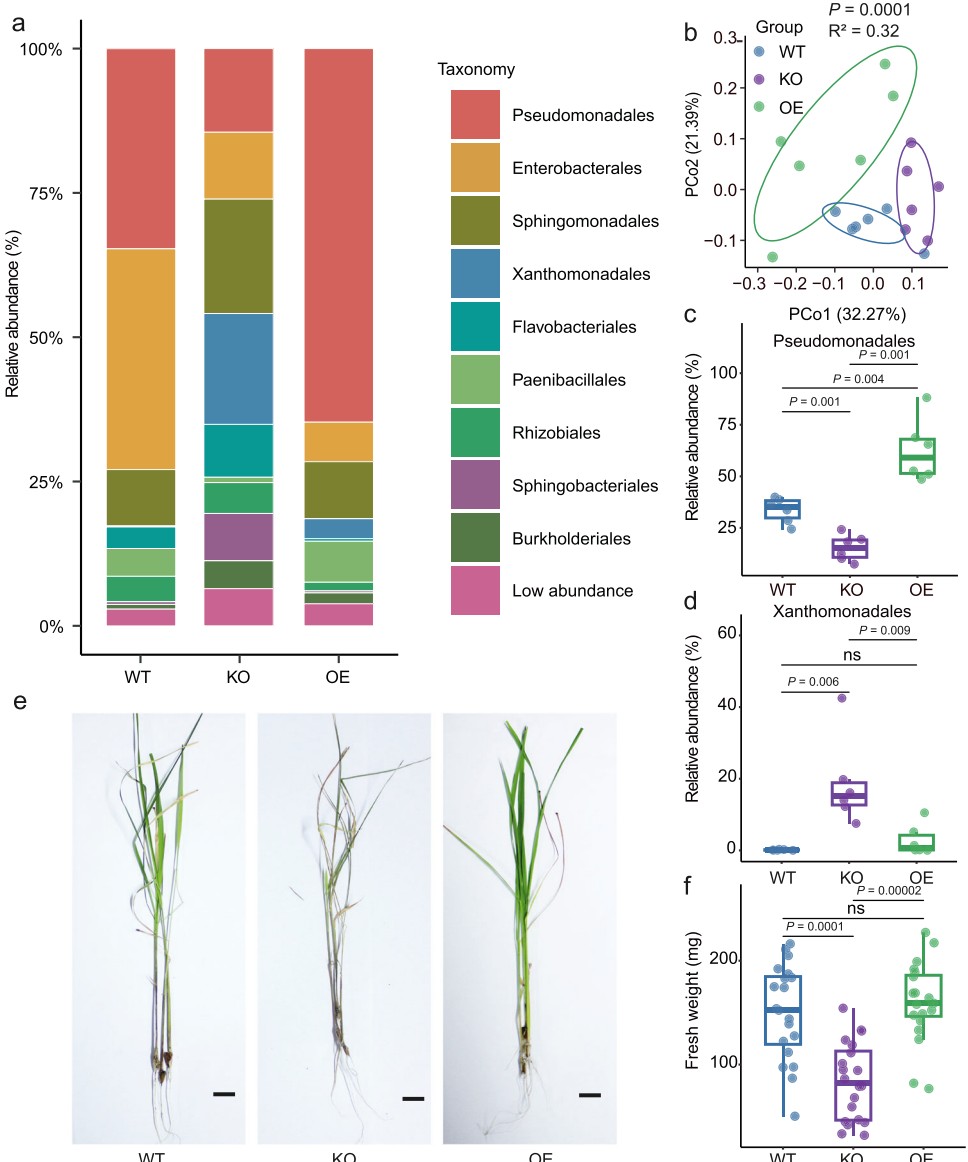

**Fig. 3 | *OsPAL02* is linked to phyllosphere microbiome assembly and has implications for plant health. a** Bacterial communities at order level in WT (wild-type, ZH11 rice variety), KO (*OsPAL02*-knockout) and OE (*OsPAL02*-overexpression) plants. **b** Unconstrained PCoA based on Bray-Curtis distances showing bacterial community clustering in WT, KO and OE plants ($P < 0.001$, $R^2 = 0.32$, *P*-value was calculated by one-way PERMANOVA). Ellipses cover 68% of the data for each group. **c**, **d** Comparison of relative abundances of Pseudomonadales (**c**) and Xanthomonadales (**d**) in WT, KO and OE phyllosphere microbiomes. The numbers of replicated samples in **a**–**d** are as follows: WT ($n = 6$), KO ($n = 6$) and OE ($n = 6$). **e**, **f** Representative images of typical phenotype traits (**e**) and fresh weight (**f**) between WT, KO and OE rice plants. The numbers of replicated samples are as follows: WT ($n = 20$), KO ($n = 20$) and OE ($n = 20$). Scale bar, 1 cm. In this figure, box plot percentiles are the same as in Fig. 1b. The *P*-values were calculated with unpaired one-way ANOVA with Tukey's HSD test ($P < 0.05$). The labels 'ns' indicate a not significant difference ($P > 0.05$). Source data are provided as a Source Data file.

with CPCoA (Supplementary Fig. 10) where 46.1% of total variance was explained by the plant genotype ($P < 0.001$, PERMANOVA). Further analysis revealed that 12 metabolites showed significant differences between WT, *OsPAL02-OE,* and *OsPAL02-KO* (Supplementary Data 6). Therein, 4-HCA showed significantly higher intensity in *OsPAL02-OE* and lower in *OsPAL02-KO* compared to WT plants (Fig. 4c), suggesting that 4-HCA is the key metabolite regulated by *OsPAL02* and may thus contribute to phyllosphere microbiome assembly. LC-MS analysis was conducted to detect 4-HCA physiological concentrations in rice leaves. As expected, the concentration of 4-HCA was significantly higher in the three *OsPAL02-OE* lines and lower in the three *OsPAL02-KO* lines when compared to WT plants (Supplementary Fig. 11, Supplementary Table 6, Supplementary Data 5). The concentration of 4-HCA was additionally determined in five Pseudomonadales-enriched *japonica*

and five Xanthomonadales-enriched *indica* varieties. We found that the concentration of 4-HCA in the analyzed *japonica* varieties was significantly higher than in the *indica* varieties (Supplementary Fig. 12, Supplementary Table 7). A correlation analysis indicated that the concentration of 4-HCA shows a positive relationship with the relative abundance of Pseudomonadales and a negative relationship with the abundance of Xanthomonadales in the leaves of the analyzed rice varieties (Supplementary Fig. 12). These observations provided further evidence that 4-HCA is the key metabolite regulated by *OsPAL02* and contributes to phyllosphere microbiome variations between *indica* and *japonica* varieties.

Apart from the altered phyllosphere microbiomes, *OsPAL02-KO* developed severe disease symptoms compared with WT and *OsPAL02-OE* after leaf cutting (Fig. 3e). This phenomenon was not observed

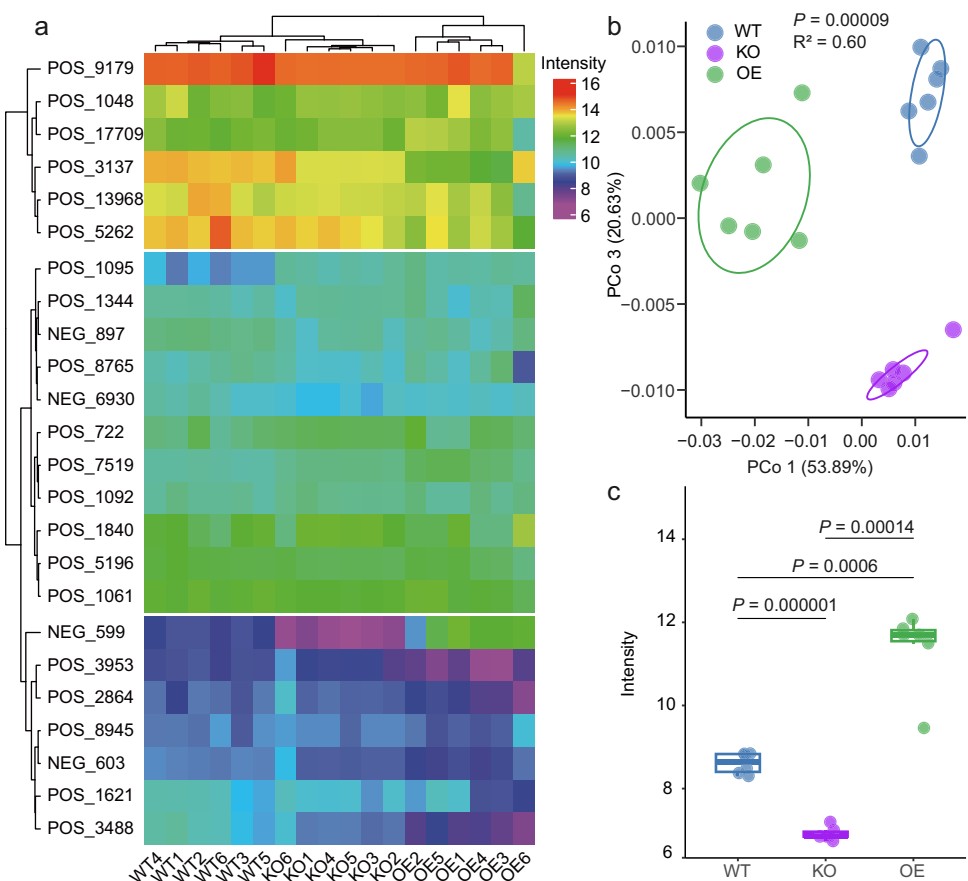

**Fig. 4 | Metabolite profiles of *OsPAL02* mutants. a** Heatmap based on intensity values of detected cinnamic acid and derivatives in different rice leaves. Clusters are shown on the left and top using Bray-Curtis distances. All cinnamic acid labels (left) are shown in Supplementary Data 6. Rice sample labels (bottom) are as follows: WT1/2/3/4/5/6 (wild-type 1/2/3/4/5/6), KO1/2/3/4/5/6 (*OsPAL02*-KO 1/2/3/4/5/6), OE1/2/3/4/5/6 (*OsPAL02*-OE 1/2/3/4/5/6). **b** Unconstrained PCoA based on Bray-Curtis distances showing leaf metabolite separation for WT, KO and OE plants ($P < 0.001$, $R^2 = 0.60$, *P*-values were calculated with one-way by PERMANOVA). Ellipses cover 68% of the data for each group. Blue, purple and green colors

represent WT, KO and OE rice plant samples, respectively. **c** Intensities (based on LC-MS/MS analyses) of 4-HCA (4-Hydroxycinnamic acid) in WT, KO and OE plant leaves, respectively. In this figure, the peak intensity values were normalized using ln transformation. Box plot percentiles are the same as in Fig. 1b. The *P*-values were calculated with unpaired one-way ANOVA with Tukey's HSD test ($P < 0.05$). Groups are abbreviated as: wild-type, WT; *OsPAL02*-knockdown, KO; *OsPAL02*-over-expression, OE. The numbers of replicated samples are as follows: WT ($n = 6$), KO ($n = 6$) and OE ($n = 6$). Source data are provided as a Source Data file.

before the leaf cutting (Supplementary Fig. 13), indicating that the absence of *OsPAL02* caused the dysbiosis in the rice phyllosphere microbiome and consequently led to the vulnerability of plants to pathogens on leaves. Unsurprisingly, the fresh weight of WT and *OsPAL02-OE* leaves was significantly higher than that of *OsPAL02-KO*; fresh weight of *OsPAL02-OE* leaves showed no significant difference to WT after leaf cutting (Fig. 3f). To determine whether disease symptoms were caused by the microbiota shift in *OsPAL02-KO*, we grew WT and *OsPAL02-KO* plants under gnotobiotic conditions and conducted the leaf cutting under controlled conditions preventing contamination. The *OsPAL02-KO* plants were indistinguishable from WT plants, and the fresh weight showed no significant differences among the three rice lines (Supplementary Fig. 14). This observation indicated that the disease symptoms in *OsPAL02-KO* might result from a microbiota dysbiosis. Combining the results of the metabolite analysis and the above assays, we proposed that OsPAL02 might be responsible for producing 4-HCA and maintaining a functional microbiota.

To validate the function of 4-HCA, rice leaves of *OsPAL02-KO* plants were sprayed with 4-HCA. As we expected, the application of 4-HCA on *OsPAL02-KO* plants resulted in a phyllosphere microbiome showing no significant difference to WT plants, and their fresh weights also did not differ (Supplementary Fig. 15, Supplementary Table 8). These results demonstrated that 4-HCA prevents dysbiosis in the

phyllosphere microbiota of *OsPAL02-KO* plants. Overall, our data suggested that OsPAL02-synthesized 4-HCA plays a key role in maintaining plant health.

## Implications of 4-HCA and members of Pseudomonadales for phyllosphere microbiome homeostasis

Although we observed that 4-HCA is involved in microbiome assembly and prevents dysbiosis in the rice phyllosphere, the underlying mechanism remained unclear. To investigate potential interactions of 4-HCA with rice leaf-associated bacteria, we isolated bacteria using the limiting dilution method[31]. Bacterial strains were isolated from pooled leaf samples of WT, *OsPAL02-OE,* and *OsPAL02-KO* plants grown under greenhouse conditions. Taxo-nomic assignments of bacterial strains were conducted by Sanger sequencing of their 16S rRNA genes, and non-redundant strains were identified based on gene similarity <97%. In total, we obtained a collection of 77 non-redundant strains that contained 14 bacterial orders, mainly Pseudomonadales (22.08% of the total non-redundant strains), Bacillales (19.48%), Enterobacterales (15.58%), Sphingomonadales (9.1%) and Flavobacteriales (7.79%). They cov-ered 62.5% of the orders (relative abundance > 0.1%) detected with amplicon sequencing analyses (Supplementary Fig. 16, Supple-mentary Data 7). The isolate collection enabled us to further study

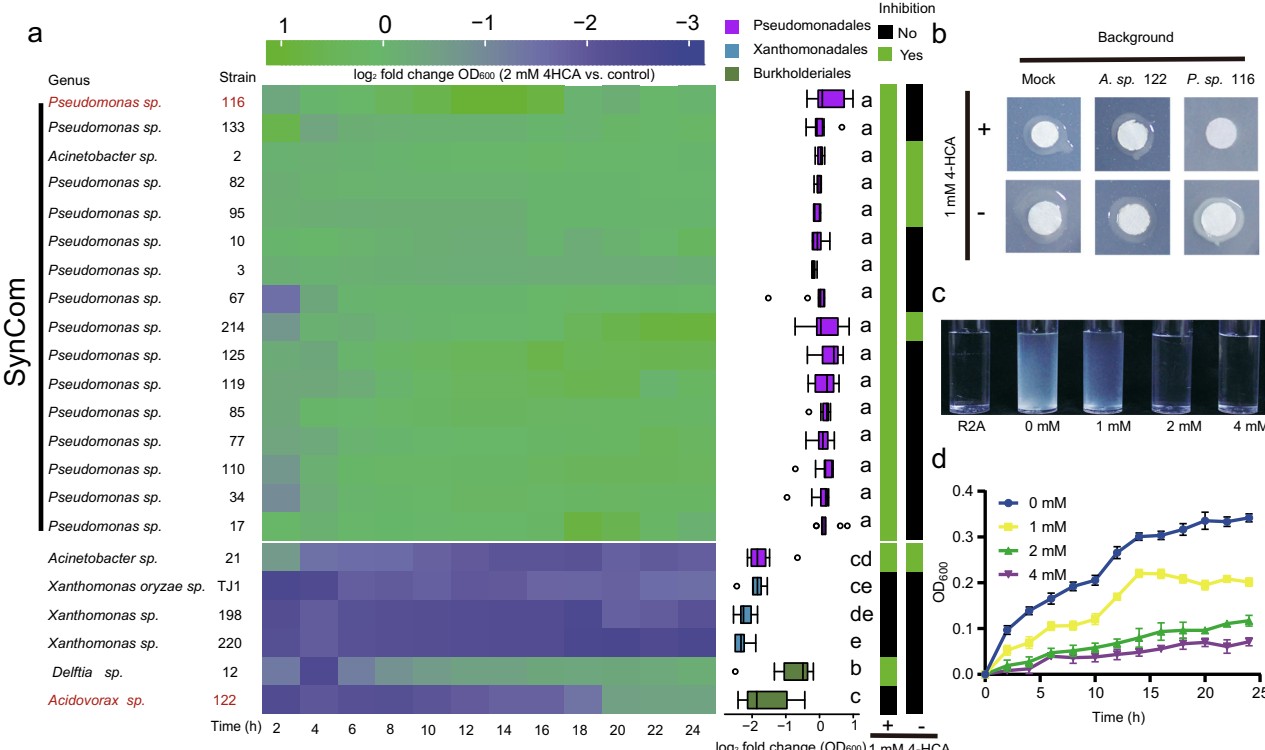

**Fig. 5 | Effects of 4-HCA on phyllosphere bacterial growth and rice pathogen TJ1. a** Left: Effects of 4-HCA on growth of 22 rice leaf-associated bacterial isolates from the orders Pseudomonadales, Xanthomonadales and Burkholderiales. The heatmap shows log$_2$ fold changes of individual isolates exposed to 2 mM 4-HCA versus control (supplemented with solvent) over 24 hours. The corresponding growth curve data are presented in Supplementary Fig. 18 and Supplementary Data 7. Middle: Statistical analysis of log$_2$ fold changes in cell density (OD$_{600}$) for the 22 isolates. Different colors represent isolates assigned to Pseudomonadales, Xanthomonadales and Burkholderiales, respectively. Box plot percentiles are the same as in Fig. 1b. Different letters indicate a significant difference according to unpaired one-way ANOVA with Tukey's HSD test ($P < 0.05$, $P$-values are listed in Supplementary Data 7, experiments for each isolate were replicated 6 times). Each

isolate's log$_2$ fold change in cell density (OD$_{600}$) of 12 time points is shown on the left. Right: In vitro inhibition of *Xanthomonas oryzae* strain TJ1 (abbreviated as TJ1) by the 22 isolates. Green or black color indicates that the respective isolate either inhibited or did not inhibit TJ1 on R2A agar with or without 1 mM 4-HCA (bottom). **b** Representative photographs of bacterial strains tested for inhibition of TJ1 in (**a**). Paper discs were inoculated with 10 μL TJ1. Presence of 1 mM 4-HCA in R2A agar is shown on the left. Background indicates isolates poured into the agar (top). Mock indicates bacterial filtrates poured into the agar. **c, d** Representative photographs (**c**) and growth curves (**d**) of TJ1 grown in different concentrations of 4-HCA. TJ1 was grown in R2A medium supplemented with 0, 1, 2, and 4 mM 4-HCA. Values are means ± SD (shown as error bars; $n = 6$ replicates). Source data are provided as a Source Data file.

interactions between 4-HCA and phyllosphere microbiota members through targeted assays and synthetic community (SynCom) approaches.

Since the disease symptoms after leaf cutting were observed not only in *OsPAL02-KO* plants but also in some WT and *OsPAL02-OE* plants (Fig. 3e), we hypothesized that the pathogenic agent might be present within the isolate collection. To identify the causal agent(s) of the disease symptoms, we tested the pathogenicity of the individual isolated strains. We inoculated single strains onto germ-free WT seedlings and monitored the plant disease occurrence. We then identified *Xanthomonas oryzae* strain TJ1 (hereafter referred to as TJ1), as a causative pathogen of bacterial leaf-blight disease on rice (Supplementary Fig. 17). We then tested the pathogenicity of TJ1 on *OsPAL02-KO* and *OsPAL02-OE* plants. As expected, the inoculation of TJ1 caused severe disease symptoms on *OsPAL02-KO*, meanwhile, *OsPAL02-OE* were less affected in terms of disease occurrence and fresh weight, compared to WT and *OsPAL02-KO*. (Supplementary Fig. 17). We also classified disease symptoms into four grades (1, healthy, to 4, dead), and showed that increasing disease severity correlated with decreasing plant fresh weight (Supplementary Fig. 17). Furthermore, we re-isolated bacteria from TJ1-inoculated WT, *OsPAL02-KO* and *OsPAL02-OE* plants. Sequencing of their 16S rRNA genes confirmed 100% sequence similarity with TJ1 (Supplementary Fig. 17). Additional isolation experiments showed that TJ1 abundance in *OsPAL02-OE* plants was significantly lower than in the *OsPAL02-KO* mutant and WT plants

(Supplementary Fig. 17). This provided further evidence that TJ1-induced disease resulted in the observed reduced average plant weight. Overall, the isolate TJ1 was confirmed to be a rice pathogen, and it was demonstrated to cause lighter disease symptoms in *OsPAL02-OE* than that observed with *OsPAL02-KO* and WT plants.

We then hypothesized that 4-HCA at certain concentrations may contribute to inhibition of TJ1 and reduce disease symptoms. We grew TJ1 in liquid culture with different concentrations of 4-HCA. We found that 4-HCA at certain concentrations can effectively inhibit TJ1 growth, and the growth of TJ1 was completely inhibited when the concentration of 4-HCA reached 2 mM (Fig. 5c, d). This result indicated that 4-HCA can directly inhibit the growth of the pathogen; the 50% inhibitory concentration (IC50) of TJ1 was 1 mM.

Next, we conducted in vitro assays to test the influence of 4-HCA on strains from Pseudomonadales, Xanthomonadales and Burkholderiales. These bacterial strains were grown in liquid R2A medium supplemented with 2 mM 4-HCA; the same concentration inhibited the growth of pathogen TJ1. We found that most of the Pseudomonadales strains grew faster in the presence of 2 mM 4-HCA, whereas all Xanthomonadales and Burkholderiales strains were significantly inhibited (Fig. 5a). Detailed analysis of growth changes (OD$_{600}$) indicated that all Pseudomonadales strains, except *Acinetobacter* sp. 21, showed a significantly higher growth rate than Xanthomonadales and Burkholderiales strains in the presence of the 2 mM 4-HCA (Fig. 5a, Supplementary Fig. 18, Supplementary Data 7). The in vitro assays clarified the effect of

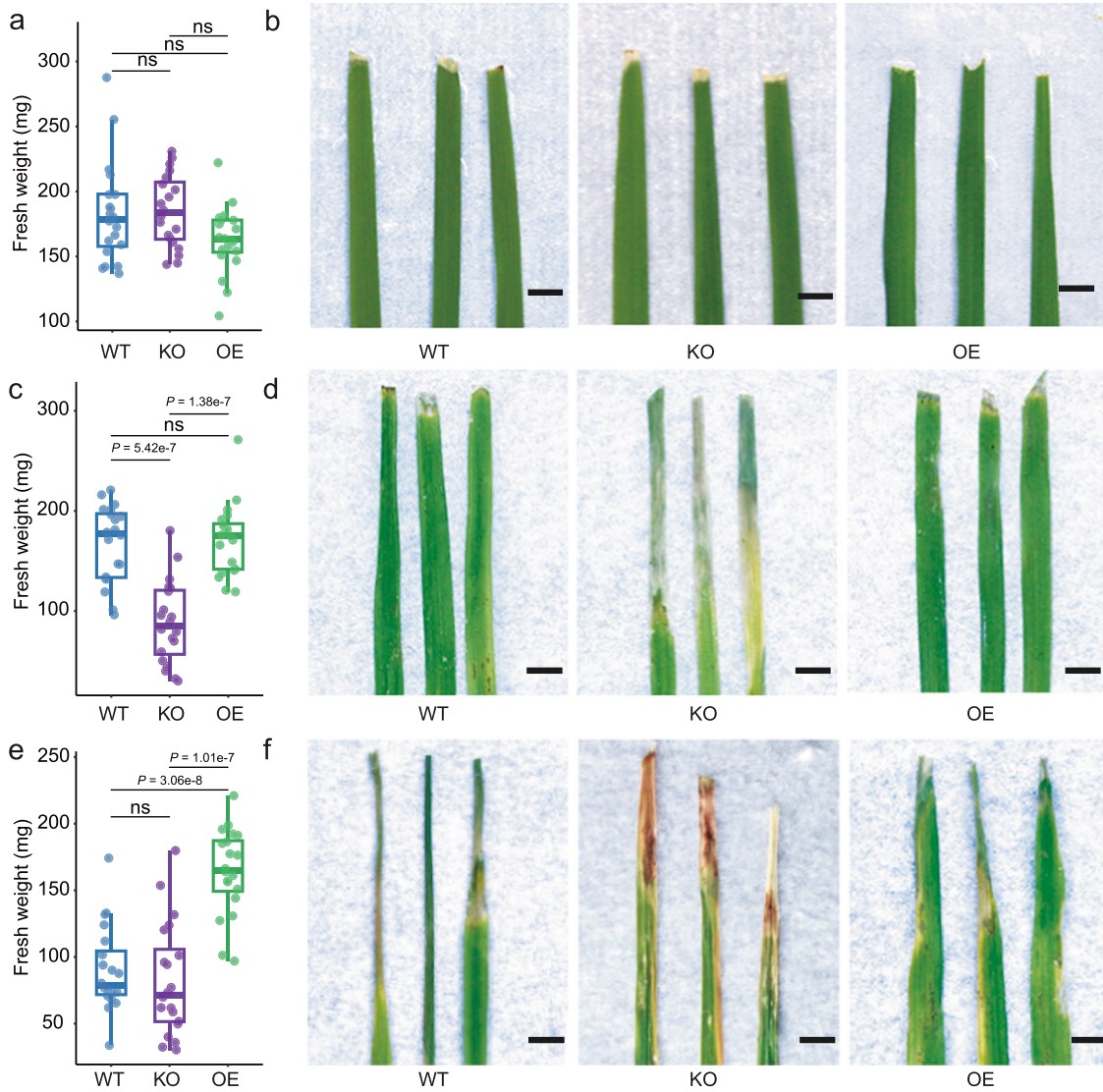

**Fig. 6 | SynCom inhibits TJ1 under 4-HCA presence in vivo. a**, **b** Fresh weight (**a**) and representative images of typical phenotypic traits (**b**) between sterile-grown WT, KO and OE rice plants treated with 10 mM MgCl$_2$. **c**, **d** Fresh weight (**c**) and representative images of typical phenotypic traits (**d**) between sterile-grown WT, KO and OE rice inoculated with synthetic community (abbreviated as SynCom) + *Xanthomonas Oryzae* TJ1 (abbreviated as TJ1). **e**, **f** Fresh weight (**e**) and representative images of typical phenotypic traits (**f**) between sterile-grown WT, KO, and OE rice plants inoculated with TJ1. Box plot percentiles are the same as in Fig. 1b.

Blue, purple and green colors represent WT, KO and OE rice plants, respectively. The *P*-values were calculated with unpaired one-way ANOVA with Tukey's HSD test (*P* < 0.05). The labels 'ns' indicate a not significant difference (*P* > 0.05). The numbers of replicated samples in each treatment are as follows: WT (*n* = 20), KO (*n* = 20) and OE (*n* = 20). Scale bar, 5 mm. Groups are abbreviated as WT (wild-type), KO (*OsPALO2*-knockdown), OE (*OsPALO2*-overexpression). Source data are provided as a Source Data file.

---

4-HCA on the growth of strains from the orders Pseudomonadales, Xanthomonadales and Burkholderiales. They also shed further light on the phyllosphere microbiome assembly discrepancies between *OsPALO2-OE*, *OsPALO2-KO*, and WT plants (Fig. 3a) and indicated a pivotal role of 4-HCA therein.

We observed severe disease symptoms in TJ1-inoculated WT plants under gnotobiotic conditions (Supplementary Fig. 17), interestingly, this phenomenon did not occur when plants were grown under greenhouse conditions, where they were exposed to various microbes (Fig. 3e, f). Therefore, we speculated that the effect of 4-HCA at least partially relies on microbiota homeostasis. We performed in vitro inhibition assays with TJ1 and 22 isolates representing leaf-associated Pseudomonadales, Xanthomonadales and Burkholderiales members, on R2A medium with and without 1 mM 4-HCA (Fig. 5a). The results revealed that under the 1 mM 4-HCA condition, all Pseudomonadales strains, including *Acinetobacter* sp. 21, strongly inhibited TJ1

(Fig. 5a). Conversely, 75% of the Pseudomonadales strains lost the ability to inhibit TJ1 when 4-HCA was absent (Fig. 5a). Within Burkholderiales, only *Delftia* sp. 12, inhibited TJ1 in the presence of 4-HCA and lost this inhibition activity when 4-HCA was absent (Fig. 5a).

Overall, 4-HCA-dependent inhibition mostly occurred among strains from Pseudomonadales. This observation also connected 4-HCA to microbial functioning in the phyllosphere. Furthermore, we designed a synthetic community (SynCom, Fig. 5a) consisting of Pseudomonadales strains, and then subjected it to specific assays to evaluate its efficacy against the pathogen TJ1. Under gnotobiotic conditions, WT, *OsPALO2-KO* and *OsPALO2-OE* plants showed no significant differences in fresh weight when treated with either 10 mM MgCl$_2$ (control, Fig. 6a, b) or the SynCom suspension (Supplementary Fig. 19). This result confirmed that the SynCom did not have a negative impact on rice plants. Then, we treated the rice plants with SynCom + TJ1 suspension and TJ1 suspension only. In WT and *OsPALO2-OE* plants,

the SynCom showed protective effects against TJ1 (Fig. 6c, d). In contrast, no protection was observed with *OsPAL02-KO* plants, as indicated by strong disease symptoms and significantly lower leaf fresh weight (Fig. 6c, d). When only TJ1 suspension was applied, WT and *OsPAL02-KO* plants both showed more pronounced disease symptoms and significantly lower leaf fresh weight compared to *OsPAL02-OE* plants (Fig. 6e, f), suggesting that without the SynCom, the protection effect against TJ1 is lost. Evidently, an interplay between *OsPAL02* and Pseudomonadales members was required for the SynCom-associated plant health maintenance. Combined with the aforementioned results, a conclusion can be drawn that OsPAL02-synthesized 4-HCA is important for the enrichment of bacterial members of the Pseudomonadales and drives phyllosphere microbiome assembly. Microbiome assembly regulated by 4-HCA-regulated is of great importance to maintaining rice phyllosphere homeostasis, which is directly connected to plant health.

## Discussion

Plants are colonized by highly diverse, tissue-specific microbial communities, the plant microbiota. The plant microbiota is known to complement host functioning and to be an important factor for enhancing resilience under abiotic as well as biotic stress. Plants can assemble and maintain certain microbiota structures, which is vital for plant health. A dynamically balanced state of the microbiota is known as 'homeostasis'[32]. Disrupting homeostasis often results in a prevalence of detrimental microbiota members, leading to shifted microbiota structures and plant disease occurrence, also known as dysbiosis[33]. It is currently sparsely understood how plants assemble and maintain a functional (homeostatic) microbiota, especially in regards to their phyllosphere[33]. Previous research has mostly focused on microbe-microbe interactions that result in homeostasis and can explain certain colonization patterns[34–36]. Here, we show that specific host-microbe interplay is involved in shaping phyllosphere bacteria prevalence in rice plants. Research conducted in the past has provided indications for active host regulation of the phyllosphere microbiota. The plant immune system was demonstrated to be required for maintaining microbiota homeostasis[37–40]. Mutations of immunity-related genes in *Arabidopsis* have been shown to result in a dysbiotic phyllosphere microbiota[27,41]. In the present study, we show that rice plants deploy a potentially conserved mechanism to shape the phyllosphere microbiome in a wide range of genotypes. Except for immunity-related mechanisms, plants can also employ metabolites as a selective force to assemble candidate microbiota members to serve their interests[17]. Various plant-specific metabolites can fulfill the purpose of maintaining microbiota homeostasis on leaves[42], or suppress bacterial pathogens[43]. Nevertheless, fundamental questions remain unaddressed related to which plant genetic components are devoted to control microbiota assembly, and how this takes place on metabolite level.

Genome-wide association studies (GWAS) have proven to be a highly useful tool to shed further light on plant-microbe interactions[28,44]. In a recent study, researchers showed that the abundance of phyllosphere microbial hubs in *Arabidopsis* correlated with multiple specialized metabolic pathways affecting both bacterial and fungal communities[19,21]. It should be highlighted the previous studies were solely based on amplicon sequencing of microbial marker genes in the phyllosphere. While such approaches are valuable, this method is also known to be prone to specific biases, because marker genes of certain microbial groups are more likely to be amplified than those of others[17]. Therefore, microbiome structures are not always accurately captured. In the present study, a large-scale phyllosphere metagenome dataset was combined with GWAS and facilitated more precise insights into associations between rice plants and their phyllosphere microbes. For one, it provided further evidence for the role of the plant immune system in shaping its microbiota, such as the MAPK signaling pathway.

More importantly, it allowed us to pinpoint specific metabolic pathways and genes which we identified as suitable targets to dissect phyllosphere microbiome assembly at molecular level. Furthermore, the high resolution of taxonomic data obtained here was advantageous to connect host genetic traits to specific bacterial taxa, which was critical to validate the effects of a distinct gene and metabolite on microbiome assembly.

The present study revealed that primarily bacteria belonging to the order Pseudomonadales are enriched by 4-HCA in the rice phyllosphere. A similar observation was obtained with poplar tree endophytes and down-regulation of cinnamoyl-CoA reductase; it caused the enrichment of 4-HCA in plant tissues and consequently increased the abundance of *Pseudomonas*[45]. This genus is a ubiquitous member of the plant microbiota[46,47]. Previous studies have shown that it is an important component of a functional microbiota that can shield off pathogens and prevent diseases in plants. *Pseudomonas* were often identified as core microorganisms associated with both the plant rhizosphere and phyllosphere[35,36,48,49], and as key players in many disease-suppressive soils[50,51]. Plants benefit from *Pseudomonas* as they can produce a wide range of antimicrobial compounds[52], induce plant systemic disease resistance[53], and establish an array of chemical dialogues with plants[37,54,55]. Our identification of plant genetic components controlling *Pseudomonas* could provide an ideal target for engineering the phyllosphere microbiota of various crop plants. We also showed that *Xanthomonas*, a highly prevalent phytopathogen, is negatively affected by 4-HCA. It is therefore very likely that plants protect themselves against pathogens by relying on their immune system as well as specific metabolites that not only allow them to directly antagonize pathogens but also to maintain homeostasis in the phyllosphere microbiota.

In recent years, plant microbiome studies have facilitated the identification of various disease-reducing or even disease-preventing microorganisms[56]. Leaves are common entry points of various, highly devastating pathogens. The present study shows that metabolite-driven regulation of the phyllosphere microbiota is a hidden mechanism to support plant hosts against pathogen attacks. Previous studies that assessed the rhizosphere microbiome have resulted in similar findings. For example, iron-mobilizing coumarins in *Arabidopsis* roots were linked to shaping the rhizosphere microbiota by inhibiting proliferation of certain *Pseudomonas* species via a redox-mediated mechanism[54]. Re-shaping of the rhizosphere microbiome by the coumarin scopoletin was shown to have consequences for plant health[57]. Similarly, the benzoxazinoid breakdown product 6-methoxy-benzoxazolin-2-one (MBOA) was found to regulate rhizosphere microbiota assembly of cereal crops[58].

The present findings related to plant metabolite-driven phyllosphere homeostasis and their implications for host health will serve as a basis for targeted breeding strategies. Previous studies have provided certain indications, mostly via correlation analyses, that plant metabolites play an important role in maintaining a functional microbiota[42]. However, it remained unresolved if the identified compounds are the main drivers of the observed structural changes of the microbial communities. In the present study, knock-out as well as over-expression constructs allowed us to specifically attribute microbiome changes to 4-HCA in the rice phyllosphere. A lack of 4-HCA was linked to dysbiosis of the phyllosphere microbiome which results in higher susceptibility of rice plants to disease. Current resistance breeding strategies known as 'R gene breeding strategies' are increasingly challenged by rapidly evolving phytopathogens. This problem could be addressed by identifying and engineering microbiome-shaping genes, which may become known as 'M gene breeding'.

## Methods

### Chemicals, reagents and analytical instruments

The chemicals, reagents and analytical instruments used in this study are listed as follows: ethanol (EtOH), sodium hypochlorite (NaOCl),

Triton X-100, methanol (MeOH), 4-hydroxycinnamic acid (4-HCA), $MgCl_2$, Graphitized carbon black (GCB), Acetonitrile (ACN) and other organic solvents was purchased from Merck (Shanghai, China); agar powder, agarose gel, PBS buffer, Kanamycin, Hygromycin, Murashige and Skoog (MS), Linsmaier–Skoog (LS), Reasoner's 2 A (R2A) and tryptic soy broth (TSB) medium were purchased from Solarbio (Beijing, China); *Bsa*I, T4 DNA Ligase, Takara Ex Taq were purchased from Takara (Beijing, China); TransScript® II All-in-One First-Strand cDNA Synthesis SuperMix for PCR, TransTaq® DNA Polymerase High Fidelity (HiFi), EasyPure® Quick Gel Extraction Kit a were purchased from TransGen Biotech (Beijing, China); FastPure Plant DNA Isolation Mini Kit was purchased from Vazyme Biotech (Nanjing, China); MagPure Soil DNA LQ Kit was purchased from Magen (Shanghai, China); TIANprep Plasmid Midi kit was purchased from TIANGEN (Beijing, China); Agencourt AMPure XP beads was purchased from Beckman Coulter (Pasadena, USA); Qubit dsDNA assay kit was purchased from Yeasen (Shanghai, China); TRIzol was purchased from Invitrogen (Carlsbad, USA).

The analytical equipment used included a UHPLC (1290 Infinity LC, Agilent Technologies, USA) coupled to a quadrupole time-of-flight mass (TripleTOF 6600, AB Sciex, USA), and a NanoDrop ND-1000 spectrophotometer (Thermo Fisher Scientific, USA).

## Rice phyllosphere metagenomes

In order to investigate potential influence of rice genotypes on the phyllosphere microbiome, we obtained shotgun-sequenced metagenomes of 110 rice varieties (three replicates for each variety). Details about this dataset were included in a dedicated data publication (Supplementary Data 1)[29]. The 110 rice varieties were selected from the Rice Diversity Panel II core collection (C-RDP-II), which can be grouped into 'japonica' (including 27 tropical *japonica*, 7 temperate *japonica*, and 2 admixed *japonica*) and 'indica' (including 32 *indica*, 22 *aus*, 2 admixed *indica*) varieties as defined by McCouch et al.[59]. All varieties were planted in Taojiang County, Hunan Province, China (28°38'09" N, 112°0'57" E). Rice leaves were harvested at the booting stage (September 5th, 2020). The samples were enriched for their bacterial fraction, metagenomic DNA was extracted and subjected to high-throughput sequencing[29]. Quality filtering of raw metagenomic data was described in the data publication[29].

## Bioinformatics analyses of the metagenomes

Quality-filtered data were subjected to the taxonomic classification using Kraken2 v2.0.9[30]. A species abundance table was generated with Bracken v2.6.0[30]. The cumulative-sum scaling normalization method was applied with R package metagenomeSeq[60]. A principal coordinates analysis (PCoA) based on Bray Curtis distances was performed using the R package vegan[61]. Alpha diversity analysis was carried out using aforementioned R package vegan. Differences in beta diversity were accessed using a pairwise Adonis test and permutational analysis of variance (PERMANOVA, 999 permutations) using R package amplicon[9]. Analysis of differential species abundance was performed using Wilcoxon rank sum and Tukey's HSD tests based on species with an average relative abundance >0.001% in the assessed rice varieties. Species were defined as significantly different abundant if the *P*-value was <0.05. Diversity and relative species abundance data were visualized by using R package amplicon. Bacterial taxonomy was visualized with GraPhlAn v.0.9.7[62]. Boxplots and scatter diagrams were visualized by using the R package ggplot2 v.2.2.1[63].

## GWAS analysis

The genome-wide association study was based on 110 C-RDP-II rice accessions represented by a 168,699-SNP dataset (filter removal MAF < 0.05) (Supplementary Data 8). This dataset was extracted from the C-RDP-II collection that encompasses a 700,000-SNP dataset[59]. We defined 'traits' as the enrichment levels of the 6,862 identified species

in the phyllosphere of the 110 rice varieties (Supplementary Data 3). Tassel5.0 (https://www.maizegenetics.net/tassel) was used for GWAS and an analysis pipeline was set up on a Linux system to analyze the 6,862 'traits' individually (scripts are available at Code Availability). The GWAS method was based on a rice genome-wide association analysis approach using a high-density SNP array[64]. We used a mixed linear model (MLM) that integrated the kinship matrix (K) with population structure (Q). We screened for significantly associated loci using the following criteria: a genomic region with ≤300 kb containing at least two significant SNPs (with a *P*-value threshold of ≤0.00001). We filtered for non-redundant loci associated with at least one enriched microorganism. The GWAS-identified loci were further analyzed for candidate genes.

## Locus clustering and pathway enrichment

GWAS-identified loci can be associated with multiple bacterial species from different orders. For each locus, we calculated the proportion of associated species from the respective order. The proportion was later subjected to clustering locus visualization. Clustering analysis based on Bray Curtis distance between all loci was conducted using R package vegan. The clustering of loci was visualized by using R package circlize[65]. For each of the locus clusters, gene annotation and Gene ID conversion were conducted using The Rice Annotation Project Database (RAPDB, https://rapdb.dna.affrc.go.jp/), and GO or KEGG enrichment analyses were conducted using R package clusterProfiler[66]. GO enrichment analysis was conducted using org.Osativa.eg.db (https://github.com/xuzhougeng/org.Osativa.eg.db) database and significant enrichment of pathways was noted if the *P*-value was <0.05 (Fisher's exact test). KEGG enrichment was analyzed using *Oryza sativa japonica* (RAPDB) KEGG Gene Database (Release 105.0 + /03-07) and significant enrichment of pathways was noted if the *P*-value was <0.05 (Fisher's exact test). GO and KEGG enrichment pathways were visualized by using ggplot2.

## Generation of CRISPR-edited and gene overexpression rice mutants

The CRISPR–Cas9 system was used to generate *OsPALO2* (Os04g0518100) mutants according to a high-efficiency monocot genome editing method[67]. Briefly, specific primers containing the *OsPALO2* target site sequences (CCATGCGCTTCACCTCGTCCAGG) were ligated into pEGCas9Pubi-H cassettes. To construct guide RNA genes, rice small nuclear RNA U6 promoter was amplified from pYLsgRNA-OsU6a vector[67] using primer pairs U6-1/ U6-2. The DNA sequence encoding the gRNA scaffold was amplified from pYLsgRNA-OsU6a vector[67] using primer pair gRNA-1/gRNA-2. The PCR product of the U6 promoter was fused with the DNA fragment encoding the gRNA scaffold by overlapping PCR using primer pair sgRNA-1/sgRNA-2. The U6 promoter–gRNA fragment was cloned into the pEGCas9Pubi-H vector to form the pEGCas9Pubi-H-Os04g0518100 construct via *Bsa*I (Takara, Beijing, China) restriction digestion followed by ligation using T4 DNA Ligase (Takara, Beijing, China) (Supplementary Fig. 20). After transformation into *Escherichia coli* DH5-alpha, the resulting constructs were purified with TIANprep Plasmid Midi kit (TIANGEN, Beijing, China) for PCR amplification using primer pairs KO-1/KO-2 and sequenced via Sanger sequencing (Sangon Biotech, Shanghai, China) (Supplementary Fig. 20). The validated constructs were stored at −20 °C for subsequent use in rice protoplast transformation.

The *OsPALO2*-overexpression constructs were generated according to a transgenic rice construction method[68]. Briefly, for the *OsPALO2* overexpression vector construct, total RNA of rice variety ZH11 leaves were extracted using TRIzol, and complementary DNA (cDNA) was synthesized from the total RNA using cDNA Synthesis SuperMix kit (TransGen Biotech, Beijing, China). The *OsPALO2* coding sequence was amplified by PCR using primer pairs PALO2-1/PALO2-2. The amplified fragment was cloned into the pEGOEPubi-H vector, which contained

the maize ubiquitin promoter and a hygromycin resistant gene, to form the pEGOEPubi-H-*OsPALO2* overexpression construct via *Bsa*I (Takara, Beijing, China) restriction digestion followed by ligation using T4 DNA Ligase (Takara, Beijing, China) (Supplementary Fig. 21). After transformation into *Escherichia coli* DH5-alpha, the resulting constructs were purified with TIANprep Plasmid Midi kit (TIANGEN, Beijing, China) for PCR amplification using primer pairs OE-1/OE-2 and sequenced via Sanger sequencing (Sangon Biotech, Shanghai, China) (Supplementary Fig. 21). The validated constructs were stored at −20 °C for subsequent use in rice protoplast transformation.

The resulting vectors were introduced into *Agrobacterium tumefaciens (strain EHA105)* and then into rice by Agrobacterium-mediated transformation[69]. The transformation was performed using a rice transformation method[69]. Briefly, positive transgenic bacteria were selected by using kanamycin for selection. Hygromycin was used to select putative transgenic plants. All the selected putatively transgenic seedlings were cultivated in the greenhouse or field for further identification.

To identify transgenic plants, genomic DNA was extracted from approximately 30 mg of leaf tissue using FastPure Plant DNA Isolation Mini Kit (Vazyme Biotech, Nanjing, China) according to the manufacturer's instructions. The isolated and purified genomic DNA was used for PCR amplification of CRISPR/Cas9 target sites or hygromycin gene using primer pairs KO2-1/KO2-2 or H-1/H-2, respectively. Amplified fragments were detected using agarose gel electrophoresis, purified using EasyPure® Quick Gel Extraction Kit (TransGen Biotech, Beijing, China) and sequenced with sanger sequencing (Sangon Biotech, Shanghai, China) to identify *OsPALO2-KO* and *OsPALO2-OE* transgenic plants. All related primers are listed in Supplementary Table 9.

### Plant material, growth conditions and treatments

Hunan Hybrid Rice Research Center (HHRRC) provided seeds of the ZH11 rice variety for the experiments and genetic transformation. ZH11 was implemented as the wild-type (WT) and to generate *OsPALO2-KO* (knockout) and *OsPALO2-OE* (overexpressing) plants. WT, *OsPALO2-OE* and *OsPALO2-KO* plants were cultured under relative humidity set at 80%, temperature at 28 °C and 13 h light cycle in a greenhouse or were grown at a field site of HHRRC in Changsha, China. Mature seeds of the WT, *OsPALO2-OE* and *OsPALO2-KO* plants were harvested, dried, and stored in a refrigerator at 4 °C.

Rice plants were grown in Taojiang soil filtrate mixed with greenhouse potting soil (PINDSTRUP substrate, Denmark; autoclaved twice) in an air-circulating greenhouse for controlled experiments. For Taojiang soil filtrate preparation, the soil was collected from the same rice field as for the metagenomic experiments, Taojiang County (28°38′09″ N, 112°0′57″ E), Hunan Province, China. The top 10–20 cm of field soil was collected and sieved (3-mm sieve) to remove rocks and other debris, dried at room temperature. Then, the soil was diluted 10-times using sterile water (1:10 w/v), followed by shaking for 20 min at 180 rpm/min and sonicated for 5 min at a frequency of 30 kHz at 4 °C. The suspensions were subjected to centrifugation for 1 min at 42 $g$, 4 °C. The supernatant was collected and used as the Taojiang soil filtrate. Greenhouse soil preparations were made by using 100 mL Taojiang soil filtrate together with 100 mL half-strength Murashige and Skoog (MS) medium solution (pH 5.8) and mixed with 200 g of greenhouse potting soil (PINDSTRUP substrate, Denmark; autoclaved twice). Rice seeds were surface-sterilized using 75% ethanol (EtOH) for 2 min and 7% sodium hypochlorite (NaOCl) supplemented with 0.2% Triton X-100 three times for 8 min, following washed with sterile water six times. Sterile seeds were further subjected to microbiota inoculation from filed soil by immersing them in Taojiang soil filtrate for 1 d at 4 °C and 4 d at 28 °C in the dark. Subsequently, all rice seeds at the early germination stage were transplanted into the pots with the aforementioned soil preparation. Each plant was watered with 5 mL of

sterile water twice a week and sterile half-strength MS solution once a week after germination. Top leaves of 3-week-old plants were cut (1 cm) using sterile scissors (surface-sterilized using 75% EtOH and washed with sterile water six times for each plant) for identifying *OsPALO2-KO* or *OsPALO2-OE* constructs 5-week-old plants were harvested for collection of phyllosphere bacteria, extraction of leaf metabolites, determination of fresh weight and other analyses.

For the complementary 4-HCA assay, WT and *OsPALO2-KO* plants were surface-sterilized, cold-stratified, germinated and transferred to the same greenhouse as mentioned before. Each 2-week-old *OsPALO2-KO* plant was sprayed with 2 mL of sterile 1 mM 4-HCA (dissolved in water). Each 2-week-old plant of WT plant was sprayed with 2 mL sterile water. Watering of the seedlings and identification of the genotype followed the same procedure as described before. 5-week-old plants were harvested for microbiome, phenotype and fresh weight analyses.

For gnotobiotic plant growth, WT, *OsPALO2-KO* and *OsPALO2-OE* seeds were surface-sterilized with the same procedure as described before, and then cold-stratified and germinated in sterile water. Subsequently, all rice seeds at early germination stage were transferred into sterile bottles with autoclaved (three times for 45 min each, with 24 h storage at room temperature in between) soil. Sterility of rice seeding was routinely monitored by plating samples of plants and culture substrate on R2A plates. Watering of seedlings and identification of the genotype were as described above, but under sterile conditions. 5-week-old plants were harvested for phenotype and fresh weight analyses.

### Phyllosphere microbiome analyses of WT and mutant plants

5-week-old plants cultivated under greenhouse conditions were harvested for profiling of the phyllosphere microbiome (six replicates per genotype, each replicate contained three plants leaves). Sample collection was conducted according to a rice phyllosphere bacteria enrichment method[29]. Total DNA was extracted using MagPure Soil DNA LQ Kit (Magen, Shanghai, China) according to the manufacturer's instructions. Quality and quantity of DNA was verified using a Nano-Drop ND-1000 spectrophotometer (Thermo Fisher Scientific, USA) and agarose gel electrophoresis, respectively. Extracted DNA was diluted to a concentration of 1 ng/μL and stored at −20 °C until further processing. PCR amplification of bacterial 16S rRNA gene fragments (V3–V4 region) was performed using Takara Ex Taq (Takara, Beijing, China) and the barcoded primers 343 F and 798 R listed in Supplementary Table 9. Amplicons were visualized using agarose gel electrophoresis and purified using Agencourt AMPure XP beads (Beckman Coulter, Pasadena, USA) twice. After purification, the DNA was quantified using Qubit dsDNA assay kit (Yeasen, Shanghai, China). Equal amounts of purified DNA were pooled for sequencing on the NovaSeq 6000 platform (Illumina Inc, USA) at Shanghai OEbiotech (Shanghai, China).

The 16S rRNA gene fragment sequences were processed using QIIME2 v.2021.4[70]. Paired-end reads were detected and the adapters were removed using Cutadapt v.2.1[71]. After trimming, paired-end reads were filtered for low-quality sequences, denoised, merged and clustered using DADA2 v.2020.2.0[72]. Amplicon sequence variants (ASVs) and a feature table were generated using QIIME2. All ASVs were annotated using the Silva v138.1 reference databases[73]. Diversity and differential abundance analyses were performed using the same methods as described above.

### Rice leaf metabolite extraction

5-week-old WT, *OsPALO2-KO* and *OsPALO2-OE* plants cultivated under greenhouse conditions were harvested for extraction of leaf metabolites (six replicates per genotype, each replicate included three plants leaves). Rice leaves were immediately frozen in liquid nitrogen and ground into fine powder with a mortar and pestle. Leaf metabolites were extracted from 80 mg of each sample using 1000 μL methanol

(MeOH)-acetonitrile (ACN)-ultrapure water (2:2:1 v/v/v). The homogenates were shaken at 200 rpm/min for 20 min, followed by sonication at a frequency of 30 kHz for 20 min. After centrifugation at 19000 $g$, 4 °C for 20 min, the supernatant was dried to remove the organic solvent using a vacuum centrifuge. The samples were redissolved using 100 µL ACN-ultrapure water (1:1 v/v) and centrifuged at 19000 $g$, 4 °C for 15 min, followed by filtering through a 0.22-µm filter before analysis with a liquid chromatography–tandem mass spectrometry (LC–MS/MS) system.

## LC-MS/MS conditions

The samples were separated by ultra-performance liquid chromatography (UHPLC) on an ACQUITY UPLC BEH C-18 column (1.7 µm, 2.1 mm× 100 mm, Waters, USA); the column temperature was 40 °C. The flow rate was set at 0.4 mL/min, and the injection volume was 2 µL. The gradient elution procedure was as follows: 0-0.5 min, 5% MeOH/H$_2$O (v/v); 0.5-10 min, linearly changed to 100% MeOH; 10-12 min, MeOH was maintained at 100%; 12.0-12.1 min, MeOH/H$_2$O linearly changed to 5%; 12.1-16 min, MeOH/H$_2$O was maintained at 5%. In MS only acquisition, the instrument was set to acquire over the m/z range 60-1000 Da, and the accumulation time for the TOF MS scan was set at 0.20 s/spectrum. In auto MS/MS acquisition, the instrument was set to acquire over the m/z range 25-1000 Da, and the accumulation time for product ion scan was set at 0.05 s/spectrum. During the whole analysis, the sample were kept in an automatic sampler at 4 °C. In order to avoid influences caused by the fluctuations of the instrument, a random sequence was used for the analysis of samples.

## Cinnamic acid and derivatives profiling

The raw MS data (wiff.scan files) were converted to MzXML files using ProteoWizard MSConvert[74]. Peak picking and peak grouping were conducted by using R package XCMS v3.20.0[75]. Annotation of adducts and calculating hypothetical masses for the group were performed with R package CAMERA (Collection of Algorithms of MEtabolite pRofile Annotation) v3.1-5[76]. In the extracted ion features, only the variables with > 50% of nonzero measurement values in at least one group were kept. Identification of cinnamic acid and derivatives was performed by comparing high-accuracy m/z value using MS/MS spectra with an in-house cinnamic acid and derivatives database established with available standards.

After normalization for total peak area intensity, clustering analysis of metabolites were performed with R package ComplexHeatmap[77]. A principal coordinates analysis (PCoA) of Bray-Curtis distances was performed using R package vegan, and a pairwise Adonis test and PERMANOVA (999 permutations) were performed using R package amplicon. Analysis of differential intensity of total peak areas was performed using Tukey's HSD test. Features were visualized by using ggplot2.

## Quantification of 4-HCA in rice leaves

Rice plants that were cultivated for five weeks under greenhouse conditions were harvested for the extraction of leaf metabolites (15 replicates per genotype, each replicate included leaves of three plants). Rice leaves were immediately frozen in liquid nitrogen and ground into fine powder with a mortar and pestle. Leaf metabolites were extracted from 0.6 g of each sample using 20 mL ACN. The homogenates were shaken at 200 rpm/min for 20 min, followed by sonication at a frequency of 30 kHz for 20 min. After centrifugation at 19000 g, 4 °C for 20 min, the supernatant was obtained and mixed with 30 mg GCB. The suspension was shaken at 200 rpm/min for 20 min and centrifuged at 19000 g, 4 °C for 15 min, followed by filtering through a 0.22-µm filter before analysis with a liquid chromatography (LC; 1290 Infinity LC, Agilent Technologies, USA) –tandem mass spectrometry (MS/MS; AB SCIEX 4500Q, AB Sciex, USA) system. The standard concentration of 4-HCA was prepared at 0.01, 0.05, 0.1,

0.5, 1, 5, 10 mg/L. The standard preparations were also mixed with 30 mg GCB, then shaken, centrifuged and filtered before LC–MS/MS experiments. The standard curve was calculated with the linear regression method by using ggplot2 (Supplementary Fig. 22). Significance of differences for 4-HCA concentrations was assessed using Tukey's HSD test. Features were visualized with ggplot2.

## Isolation of rice leaf bacteria

Rice plants cultivated under the aforementioned greenhouse conditions were used for bacterial isolations via the limiting dilution method[31]. To obtain a collection of representative rice leaf colonizing bacteria, we isolated bacteria from pooled WT, *OsPAL02-KO* and *OsPAL02-OE* leaves. Plants were harvested after five weeks. The leaves of three plants of each sample type were pooled to increase diversity of present bacteria. Rice leaves were washed twice in sterile PBS buffer (0.02 M, pH 7.0) on a shaker for 5 min at 180 rpm/min to remove small particles and loosely attached microbes. The leaves were ground into a homogeneous suspension using 10 mM sterile MgCl$_2$, followed by sonication for 5 min at a frequency of 30 kHz. Homogenized leaves were left to sediment for 15 min and the supernatants were diluted, distributed and cultivated in 96-well microtiter plates in 1:10 tryptic soy broth (TSB)-water (v/v) and 1:2 Reasoner's 2 A (R2A)-water (v/v) for 20 d at room temperature. Then bacteria were purified by continuous streaking. Cultivated bacteria were identified by Sanger sequencing with 27 F and 1492 R primers (Supplementary Table 9) as well as used for the preparation of glycerol stocks for the culture collection. Taxonomic identity was assessed using NCBI BLASTN (https://blast.ncbi.nlm.nih.gov/Blast.cgi). Non-redundant bacterial isolates were selected by 16S rDNA sequence homology <97%. The taxonomic profiling of non-redundant bacterial isolates was visualized with GraPhlAn.

## Screening for opportunistic rice pathogens

To remove seed-epiphytic microbes, ZH11 seeds were subjected to surface-sterilization as described above. The resulting seeds were placed on TSB agar plates (ten seeds per plate) to confirm that their surface is epiphyte-free by checking for the absence of visible colony growth surrounding them (24 h, 28 °C). Individual bacterial strains from the culture collection were picked from R2A agar plates and suspended in 10 mM MgCl$_2$. The final OD$_{600}$ values of bacterial suspensions were adjusted to 0.01. Subsequently, ten sterilized rice seeds were subjected to bacterial inoculation with the individual isolates by soaking them in 20 mL of bacterial cell suspension in a 9-cm (diameter) Petri dish at 28 °C until their germination. Ten seeds at the early germination stage were transplanted into a 20-cm (height) sterile glass bottle with a filter and permeable membrane containing 200 mL of Linsmaier–Skoog (LS) medium solidified with 0.8% agar. For each treatment, 20 µL bacterial cell suspension was applied directly to each leaf of the one-week-old seedlings. They were incubated in a plant growth incubator (28 °C, 80% relative humidity, 13-h photoperiod). All seedling leaves were cut 1 cm from the top after 2 weeks. The phenotype, disease index and fresh weight of the seedlings were monitored and measured for a total of 3 weeks. Only *Xanthomonas oryzae* sp. TJ1 (TJ1) showed disease symptoms from all tested bacteria. The phenotype assay with *OsPAL02-KO* and *OsPAL02-OE* plants inoculated with TJ1 was conducted in the same way as described above. Bacteria were also re-isolated from the inoculated WT, *OsPAL02-KO* and *OsPAL02-OE* plants. TJ1 was identified by Sanger sequencing with 27 F and 1492 R primers (Supplementary Table 9). Bacterial colonization was quantified by plate counting. All isolates were tested in triplicates with three technical repeats.

## Effect of 4-HCA on leaf bacterial growth

The main product of OsPAL02 is 4-HCA (4-hydroxycinnamic acid), therefore the effects on the growth of leaf bacteria were tested using in vitro bioassays with pure 4-HCA[14]. Single colonies were picked from

R2A agar plates and grown overnight in R2A medium on a shaker at 180 rpm/min, 28 °C. Overnight cultures were used to collect bacteria with centrifugation at 600 $g$, 4 °C for 10 min. The collected bacteria were resuspended to $OD_{600} = 0.01$ using R2A medium, followed by addition of sterile 1 M 4-HCA solution (dissolved in ethanol, and filtered through a 0.22-μm filter) to reach different concentrations. The same amount of the solvent EtOH (filtering through a 0.22-μm filter) was added as control. The effect of 4-HCA on the opportunistic pathogen's (TJ1) growth was assessed by using R2A medium supplemented with 0, 1, 2 or 4 mM 4-HCA. We found that R2A medium supplemented with 2 mM 4-HCA completely inhibited TJ1. Therefore, this concentration of 4-HCA was used to assess growth effects on isolated members of Pseudomonadales, Xanthomonadales and Burkholderiales in the culture collection. After supplementation with 4-HCA or solvent, 200 μl of each microbial suspension was aliquoted into clear 96-well flat-bottom, polystyrene tissue culture plates. The plates were incubated at 28 °C on an orbital shaking platform at 80 rpm/min. The optical density ($OD_{600}$) was measured every 2 hours at 600 nm and monitored over 24 hours (Supplementary Data 7). All isolates were tested in triplicates with three technical repeats.

### Assessment of TJ1 inhibition by leaf bacteria

All isolates belonging to Pseudomonadales, Xanthomonadales and Burkholderiales were screened for inhibition of TJ1 using a binary interaction assessment method[27]. Bacterial isolates were individually cultured on R2A plates at 28 °C. A sterile 1-μL plastic loop with collected bacteria was suspended in 3 mL R2A medium. To incorporate bacteria into the medium, 2.6 mL of bacterial suspension were added to 40 mL of molten R2A-agar or R2A-agar supplemented with 1 mM 4HCA pre-cooled to 42 °C, gently mixed and then poured into Petri dishes (9 cm). A mock treatment was prepared with 2.6 mL filtrate of the same bacterial suspension, through a 0.22-μm filter. Single colonies of TJ1 were picked from R2A agar plates and grown overnight in R2A medium on a shaker at 180 rpm/min, 28 °C, followed by the collection of bacteria via centrifugation at 600 g, 4 °C for 10 min. For the inhibition target, TJ1 suspension was adjusted to $OD_{600} = 0.1$ using 10 mM sterile $MgCl_2$. A total of 10 μL TJ1 suspension was spotted onto paper discs at the agar plates as well as, 10 μL filtrate of TJ1 suspension, which was filtered through a 0.22-μm filter, as control. All isolates were tested in triplicates with three technical repeats, and their inhibition ability was recorded after 2d at 28 °C.

### Synthetic community assembly and plant leaf inoculation

A synthetic community (SynCom) was designed to contain Pseudomonadales isolates on the basis of the observed improved growth when exposed to 4-HCA (shown in Fig. 5a) and prepared using a SynCom preparation method[41]. Sixteen bacterial isolates and TJ1 were individually grown on R2A agar plates, and then single colonies were re-streaked on fresh R2A plates and grown at 28 °C. A sterile 1-μL plastic loop with material of each isolate was used to resuspend them in 1 mL of sterile 10 mM $MgCl_2$. Tubes containing the resuspended bacteria were vortexed for 5 min, followed by sonication for 2 min at a frequency of 30 kHz to disperse cell aggregates. The SynCom inoculum was obtained by combining 16 bacterial isolates in equal suspension volumes. The optical density ($OD_{600}$) of the SynCom inoculum and TJ1 was adjusted to 0.02.

For plant inoculation, rice seeds of WT, *OsPAL02-KO* and *OsPAL02-OE* were surface-sterilized as described above. Each sterilized seed was germinated for 5 d at 28 °C with 800 μL of the SynCom or sterile 10 mM $MgCl_2$ (as control) in a Petri dish, followed by direct application of 50 μL TJ1 or sterile 10 mM $MgCl_2$ (as control) to each seed. Then rice seedlings were transplanted into sterilized bottles containing LS agar. They were incubated in a plant growth incubator (28 °C, 80% relative humidity, 13-h photoperiod). Leaves were cut from 3-week-old plants using sterile scissors (surface-sterilized using 75%

EtOH and washed with sterile water six times for each plant) to identify their genotype. After 2 weeks, the phenotype of each plant was recorded, and their fresh weight was measured. Different plant genotypes were grown in the same bottle to minimize bottle variations, and treatments were blinded after inoculum preparation to avoid unconscious biases.

### Reporting summary

Further information on research design is available in the Nature Portfolio Reporting Summary linked to this article.

## Data availability

Raw sequence data (16S rRNA gene fragment sequencing) generated in this study have been deposited in the Genome Sequence Archive of the BIG Data Center[78], Chinese Academy of Sciences under accession PRJCA016320. The phyllosphere metagenomes of 110 rice genotypes were deposited in the European Nucleotide Archive (ENA) database under the project PRJEB45634. The 4-HCA-associated pathway is available under the pathway map "dosa00940" in the KEGG Pathway database [https://www.kegg.jp/pathway/dosa00940]. Taxonomic classifications of shotgun-sequenced metagenomes used in this study are available in the MiniKraken2_v1_8GB database [https://ccb.jhu.edu/software/bracken/]. Taxonomic classifications of 16S rRNA gene fragments used in this study are available in the Silva v138.1 reference database [https://www.arb-silva.de/documentation/release-1381/]. Rice gene annotations used in this study are available in the Rice Annotation Project Database (RAPDB) [https://rapdb.dna.affrc.go.jp/]. GO function annotations used in this study are available at Github [https://github.com/xuzhougeng/org.Osativa.eg.db]. All isolated bacterial strains and transformed rice lines were deposited in the State Key Laboratory of Hybrid Rice and the Institute of Plant Protection of Hunan Academy of Agricultural Sciences (Changsha, China). Other data generated in this study are provided in the Supplementary Data files. Source data are provided with this paper.

## Code availability

Scripts employed in the computational analyses are available at Github [https://github.com/kanghouxiang105/micro-GWAS] and Zenodo [https://zenodo.org/records/10115039][79].

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

## Acknowledgements

P.S. and Y.L. were supported by a grant from the National Key R&D Program of China (2022YFD1400700); H.K. was supported by grants from the National Key R&D Program of China (2021YFC2600400) and the National Natural Science Foundation (32261143468); P.S. was additionally supported by a grant from the Key R&D Program of Hunan Province (2022NK2014); D.Z. was supported by a grant from the LONGPING HIGH-TECH INDUSTRY ZONE (LongPing committee issue 2022-18). We appreciate the support of Bin Liu and Junliang Zhao from the Rice Research Institute, Guangdong Academy of Agricultural Sciences, China for providing the RDP-II seeds used in this study. We also want to thank OE Biotech Co., Ltd (Shanghai, China) for their technical support during high-throughput sequencing experiments.

## Author contributions

T.C., D.Z., Y.L., and P.S. conceived the idea for the study. P.S. and Q.P. performed the bioinformatic analyses and visualized the data. H.K. conducted the GWAS analyses. T.C., P.S., Q.P., and H.K. analyzed the data. W.A.W. conducted formal data analysis and review. G.B. conducted review. P.S., Q.P., Z.L., and J.M. conducted the bacterial strain isolation, rice mutant construction and the microbiological experiments. The manuscript was written by T.C., P.S., Q.P., and H.K. All authors read and approved the final manuscript.

## Competing interests

The authors declare no competing interests.
