## [Peer Review File · Nature Communications]

Microbiome homeostasis on rice leaves is regulated by a precursor molecule of lignin biosynthesisReviewers' Comments:

Reviewer #1:

Remarks to the Author:

This manuscript reports on an important role of 4-hydroxycinnamic acid (4-HCA) in shaping the rice phyllosphere microbiome. The authors started with GWAS of INDICA and JAPONICA rice varieties and subsequently correlated the genotype-dependent phyllosphere microbiome with some metabolic pathways in the plants. The authors further showed a causative relation between 4-HCA and the abundance of Pseudomonadales in the phyllosphere, by using transgenic plants with the gene knockout or overexpression of OsPAL2, which produces 4-HCA, as well as by using commercial 4-HCA. Both 4-HCA and the isolated Pseudomonadales strains were shown to suppress a pathogen in the rice phyllosphere, thus further explaining the importance of 4-HCA to the rice phyllosphere microbiome. This study is highlighted by utilizing GWAS to identify key factors for the rice phyllosphere microbiome. The different effects of 4-HCA on Pseudomonadales and the isolated pathogen add interesting new insights into microbiome regulation by plant metabolites.

I have only some minor concerns/suggestions:

1. The title emphasizes lignin, but the results suggested a direct link between the phyllosphere microbiome and 4-HCA, whereas lignin was not investigated. I would suggest revising the title, since it is somewhat misleading.
2. Words in some figures are too small or blur to be read, e.g., Figure 2.
3. Ethanol was used as the solvent control for testing the effects of 4-HCA on bacteria culture (Line 787), but water was used as the solvent control when spraying 4-HCA to plants (Line 660). This difference should be justified.
4. Lines 527-531 about the phyllosphere microbiome samples, it is unclear whether the leaves were surface-sterilized or just washed briefly.
5. The effects of 4-HCA were tested at the concentrations of 1 or 2 mM. Are they the physiological concentrations in rice leaves? Related to this question, the effects of 4-HCA on Pseudomonadales bacteria were tested in growth medium. Did the KO mutant or OE plants show different Pseudomonadales colonization rates compared to the wild type plants under gnotobiotic conditions?
6. The GWAS of INDICA and JAPONICA led to the discovery of 4-HCA's role in the phyllosphere microbiome. Do the different varieties accumulate different levels of 4-HCA?
7. Line 431, "... a conserved mechanism...". I know 4-HCA had been reported with antimicrobial effects (e.g., Cho et al., 1998), but was its effects on phyllosphere microbiome reported in other plant species?
8. Line 451, "it provided further evidence for the role of the plant immune system in shaping its microbiota". This statement is confusing. Is 4-HCA generally considered as part of the plant immune system?
9. The discussion emphasizes engineering microbiome-shaping genes. It may be better to explain how OsPAL2 was chosen from a group of candidate genes (Lines 219-236) for further studies, in addition to Line 244 that says "we performed gene-editing to mutate a host gene ...".

Reviewer #2:

Remarks to the Author:

In this manuscript, Su et al. addressed the mechanisms by which rice plants modulate their leaf microbiome. The authors used 110 rice accessions, quantified microbial strains by metagenomics, performed a genome wide association study and identified a number of loci associated with microbial diversity. They then focused on one candidate gene and proposed that genetic variation in OsPAL02 gene underlies the variation in microbial composition. They showed that overexpression of PAL02 affects the microbiome in an opposite direction than knockout and that loss of function leads to disease symptoms. They identified 4-hydroxycinnamic acid as the responsible metabolite and could chemically complement the gene knockout. They isolated bacteria from the leaves and in SynCom experiments showed that *Pseudomonas* strains that profit from 4-HCA protect plant from the pathogenic action of a *Xanthomonas oryzae* strain TJ1.

Altogether, this is a great amount of work, some very good solid data. The unbiased GWAS approach is very commendable, the data will be helpful to identify further genes/metabolites contributing to control of microbiome assembly. Also the work on the candidate gene is solid, the conclusions on involvement of 4-HCA well justified.

However, the conclusions on PAL02 underlying the variation between indica and japonica needs a bit more work. As described, the experiments with the transgenic plants are technically not adequate. It seems that the authors used single KO and OE line, which is not acceptable. For the KO, multiple lines have to be used and/or the mutant has to be complemented. For OE plants at least 3 independent lines have to be analysed. Actually, by doing the complementation, the authors could prove their claim on PAL02 variation being responsible for the difference between japonica and indica, in that they complement the KO with the different haplotypes.

All in all, this is a very good study that provides very good advance in knowledge on microbiome assembly in rice. For me, the only limitation of the manuscript is that the authors did not relate the results with the publications on effects of lignin manipulation on microbiome, such as Becker et al. 2016, PNAS. It would also be very interesting and simple to test microbiome composition of *Arabidopsis* mutants, such as the ref3. This would allow to assess how general is the described activity of 4-HCA.

Reviewer #3:

Remarks to the Author:

Su and co-workers presented an investigation of the molecular basis regulating host-microbe interactions in the leaves of the staple crop rice. Combining an innovative genome-wide association study, using microbiota information as an "external" plant trait, with plant functional genomics, metabolomics and microbial inoculation they concluded that a precursor of plant lignin is required for the assembly of the endogenous microbiota which, in turn, fends off opportunistic pathogens in rice leaves. The manuscript reads well and, almost invariably, the conclusions represent an accurate interpretation of the data generated. What makes this story standing, in my vision, is the comprehensive and multidisciplinary approach: from gene identification, to validation and, perhaps most importantly, an effort to establish a clear causal relationships between plant genetic variation and configurations of the microbiota. For these reasons, I am confident this manuscript may represent a reference for future investigations of the plant microbiota in crop plants.

I attach a number of, relatively minor, questions and suggestions for the authors in the commented version of the manuscript.

Response to reviewer's comments

Reviewer #1 (Remarks to the Author):

This manuscript reports on an important role of 4-hydroxycinnamic acid (4-HCA) in shaping the rice phyllosphere microbiome. The authors started with GWAS of INDICA and JAPONICA rice varieties and subsequently correlated the genotype-dependent phyllosphere microbiome with some metabolic pathways in the plants. The authors further showed a causative relation between 4-HCA and the abundance of Pseudomonadales in the phyllosphere, by using transgenic plants with the gene knockout or overexpression of OsPAL02, which produces 4-HCA, as well as by using commercial 4-HCA. Both 4-HCA and the isolated Pseudomonadales strains were shown to suppress a pathogen in the rice phyllosphere, thus further explaining the importance of 4-HCA to the rice phyllosphere microbiome. This study is highlighted by utilizing GWAS to identify key factors for the rice phyllosphere microbiome. The different effects of 4-HCA on Pseudomonadales and the isolated pathogen add interesting new insights into microbiome regulation by plant metabolites.

I have only some minor concerns/suggestions:

1. The title emphasizes lignin, but the results suggested a direct link between the phyllosphere microbiome and 4-HCA, whereas lignin was not investigated. I would suggest revising the title, since it is somewhat misleading.

We would prefer to keep the original due to two reasons. For one, it would put our work in a larger context. The second is that our GWAS not only provided indications for the involvement of OsPAL02 in microbiome assembly, but also of other genes in the lignin biosynthesis pathway (they are highlighted in Fig. 2). We provided evidence for the importance of 4-HCA in this study and it is likely that future studies will confirm it for the other identified genes/enzymes.

2. Words in some figures are too small or blur to be read, e.g., Figure 2.

We improved the quality of Figure 2 in the revised manuscript. Additionally, we uploaded high-resolution PDF versions for the resubmission.

3. Ethanol was used as the solvent control for testing the effects of 4-HCA on bacteria culture (Line 787), but water was used as the solvent control when spraying 4-HCA to plants (Line 660). This difference should be justified.

The 4-HCA solutions mentioned in Line 787 and Line 660 (in the first submission) was prepared in two separate assays. In Line 787, 4-HCA was added to culture medium from a 1 M stock solution. We used the stock solution instead of directly adding 4-HCA into medium because the stock solution was more convenient to be used during the culture medium preparation. The final concentration was 1 mM 4-HCA, which means that the stock solution containing ethanol was diluted by 1:1000. In Line 660, the water-based solution with 4-HCA for leaf spray application was

prepared by directly adding the compound into water. In the revised manuscript, we highlight it in the text accordingly.

4. Lines 527-531 about the phyllosphere microbiome samples, it is unclear whether the leaves were surface-sterilized or just washed briefly.

We collected bacterial epiphytes, not endophytes from the phyllosphere. We did not conduct surface-sterilization. The processing details are as follows: The detached leaves were immediately wrapped with a sterilize gauze to minimize the leakage of leaf tissue fluid, which might contaminate the leaf samples with plant organelles and microbial endophytes. They were kept in cooling boxes at 4 °C. After leaf sampling, a total of 5 g leaf material from each rice cultivar was used to enrich bacteria from the plant phyllosphere. The leaf samples for each replicate were transferred into a 250-mL conical flask containing 100 mL sterile PBS buffer (0.02M, pH 7.0) and 100 μ L Tween-30. The flask was placed in a shaker for 1h set at 200 rpm/min and then sonicated for 5min at a frequency of 30kHz, while a temperature of 4 °C was maintained. After sonication, the leaves were recycled and treated with the same procedure two more times to ensure that the bacterial cells were thoroughly washed off from the leaf surface. The suspensions from the washing steps were pooled together and subjected to centrifugation (1,500 rpm/min, 1min, 4 °C). The supernatant was then collected and again subjected to centrifugation (12,000 rpm/min, 15min, 4 °C). The pellets obtained after the second centrifugation were stored at -80 °C before further use. The details of phyllosphere microbiome sampling are described in a previous study, and we have cited the article in line 579 in the revised manuscript.

5. The effects of 4-HCA were tested at the concentrations of 1 or 2 mM. Are they the physiological concentrations in rice leaves?

We used 1 or 2 mM to test the effects of 4-HCA because these concentrations were found to have 50% and 100% inhibitory rates on the growth of TJ1 in cultivation broth, respectively. As to the physiological concentrations in rice leaves, we conducted LC/MS assays with the WT rice ZH11 variety used for mutant lines. The concentration was 2.749 mg/g in rice leaves (Fig. S11, Table S15), which corresponds to 16.76 mM. This is substantially higher than the concentrations we used in the assay, but it has to be noted that we do not have data on the concentration of 4-HCA on the leaf surface.

Related to this question, the effects of 4-HCA on Pseudomonadales bacteria were tested in growth medium. Did the KO mutant or OE plants show different Pseudomonadales colonization rates compared to the wild type plants under gnotobiotic conditions?

We assessed colonization rates using the Pseudomonadales SynCom to inoculate wild type plants, KO mutant and OE plants under gnotobiotic conditions. 5-week-old-plants were used to re-isolate and quantify them using plate counting. The results showed that Pseudomonadales colonization was not significantly different between KO mutant, OE plants and wild type plants (shown in Figure 1 below). We assume that Pseudomonadales colonization will remain on a stable level when there are no other competitors. This result does not impact the main results, so we did not include it in this article.

Figure 1. Pseudomonadales colonization of WT, KO, and OE plants under gnotobiotic conditions.

6. The GWAS of *INDICA* and *JAPONICA* led to the discovery of 4-HCA's role in the phyllosphere microbiome. Do the different varieties accumulate different levels of 4-HCA?

We conducted additional experiments to explore this. For this, we determined concentrations of 4-HCA in five Pseudomonadales-enriched *JAPONICA* and five Xanthomonadales-enriched *INDICA* varieties and included the results in line 323. We found that the concentration of 4-HCA in *JAPONICA* varieties was significantly higher than in *INDICA* varieties (Fig. S12, Table S15). And correlation analysis indicated that the concentration of 4-HCA shows a positive relationship with the relative abundance of Pseudomonadales and a negative relationship with the relative abundance of Xanthomonadales in the leaves of the 10 tested rice varieties (Fig. S12).

7. Line 431, "... a conserved mechanism...". I know 4-HCA had been reported with antimicrobial effects (e.g., Cho et al., 1998), but was its effects on phyllosphere microbiome reported in other plant species?

We argue that this is a conserved mechanism because 4-HCA is a common compound in plant tissues. In our resubmission, we have toned down and added "a potentially conserved mechanism...." in line 472.

8. Line 451, "it provided further evidence for the role of the plant immune system in shaping its microbiota". This statement is confusing. Is 4-HCA generally considered as part of the plant immune system?

The GWAS provided associations indicating the involvement of the plant immune system in shaping its microbiota, in addition to 4-HCA related genes. We replaced "For one, it provided further evidence for the role of the plant immune system in shaping its microbiota" with "For one, it provided further evidence for the role of the plant immune system in shaping its microbiota, such as the MAPK signaling pathway" in **line 493** of the revised manuscript.

9. The discussion emphasizes engineering microbiome-shaping genes. It may be better to explain how OsPAL2 was chosen from a group of candidate genes (Lines 219-236) for further studies, in addition to Line 244 that says "we performed gene-editing to mutate a host gene ...".

OsPAL2 was selected as the target gene due to the strong association with the highly abundant order Pseudomonadales (p value: 1.33E-09) and because there are no paralogous genes in locus 158 and other loci identified by GWAS (Fig. 2C, D). We have added this information in **line 253** of the revised manuscript.

Reviewer #2 (Remarks to the Author):

In this manuscript, Su et al. addressed the mechanisms by which rice plants modulate their leaf microbiome. The authors used 110 rice accessions, quantified microbial strains by metagenomics, performed a genome wide association study and identified a number of loci associated with microbial diversity. They then focused on one candidate gene and proposed that genetic variation in OsPAL2 gene underlies the variation in microbial composition. They showed that overexpression of PAL2 affects the microbiome in an opposite direction than knockout and that loss of function leads to disease symptoms. They identified 4-hydroxycinnamic acid as the responsible metabolite and could chemically complement the gene knockout. They isolated bacteria from the leaves and in SynCom experiments showed that Pseudomonas strains that profit from 4-HCA protect plant from the pathogenic action of a Xanthomonas oryzae strain TJ1. Altogether, this is a great amount of work, some very good solid data. The unbiased GWAS approach is very commendable, the data will be helpful to identify further genes/metabolites contributing to control of microbiome assembly. Also the work on the candidate gene is solid, the conclusions on involvement of 4-HCA well justified.

However, the conclusions on PAL2 underlying the variation between indica and japonica needs a bit more work. As described, the experiments with the transgenic plants are technically not adequate. It seems that the authors used single KO and OE line, which is not acceptable. For the KO, multiple lines have to be used and/or the mutant has to be complemented. For OE plants at least 3 independent lines have to be analysed. Actually, by doing the complementation, the authors could prove their claim on PAL2 variation being responsible for the difference between japonica and indica, in that they complement the KO with the different haplotypes.

All in all, this is a very good study that provides very good advance in knowledge on microbiome assembly in rice. For me, the only limitation of the manuscript is that the authors did not relate the results with the publications on effects of lignin manipulation on microbiome, such as Becker

et al. 2016, PNAS. It would also be very interesting and simple to test microbiome composition of Arabidopsis mutants, such as the ref3. This would allow to assess how general is the described activity of 4-HCA.

1. As described, the experiments with the transgenic plants are technically not adequate. It seems that the authors used single KO and OE line, which is not acceptable. For the KO, multiple lines have to be used and/or the mutant has to be complemented. For OE plants at least 3 independent lines have to be analyzed.

We conducted a phyllosphere microbiome analysis with two more KO and OE lines that were available in our lab. They are designated as Koline2, Koline3, Oeline2 and Oeline3 in the Supplementary Data. The results were presented in Figure S9. A description of the results was added in line 284 of the revised manuscript. The concentrations of 4-HCA in the added lines and lines used in our first submission (named Koline1 and Oeline1) were also presented in Figure S11. The description of these results was added in line 320 of the revised manuscript. Overall, the additionally obtained findings are in line with the mutants that we initially used. The relative abundance of Pseudomonadales showed a significant decrease in Koline2 and Koline3 and a significant increase in Oeline2 and Oeline3, compared to WT plants (Fig. S9D). The relative abundance of Xanthomonadales showed a significant increase in Koline2 and Koline3 compared to WT, Oeline2, Oeline3, respectively (Fig. S9E). The concentrations of 4-HCA in the three KO lines were found to be significantly lower than that of WT plants. The concentrations of 4-HCA in the three OE lines were found to be significantly higher than that of WT plants (Fig. S11).

2. Actually, by doing the complementation, the authors could prove their claim on PAL02 variation being responsible for the difference between japonica and indica, in that they complement the KO with the different haplotypes.

We did not conduct a complementation of OsPAL02 gene in the KO lines. Instead, in addition to the analysis of the OsPAL02 variations between *JAPONICA* and *INDICA* cultivars (in our first submission), we further analyzed OsPAL02 haplotype 1 which mainly exists in *INDICA* cultivars, and haplotype 5 which mainly exists in *JAPONICA* cultivars (presented in our resubmission). We found that haplotype 1 contains glycine at position no. 134, but haplotype 5 contains cysteine at the same position of the OsPAL02 protein sequence (Fig. S7). We argue that this variation could cause catalytic activity difference between *JAPONICA* and *INDICA* varieties. We also think that this analysis provides a basis for a better understanding of how OsPAL02 variations influence 4-HCA production between *JAPONICA* and *INDICA* plants. In addition, two more OE lines were added in the revised manuscript; they support the role of OsPAL02 in the *JAPONICA* haplotype (the wild type ZH11 is a *JAPONICA* variety).

3. For me, the only limitation of the manuscript is that the authors did not relate the results with the publications on effects of lignin manipulation on microbiome, such as Becker et al. 2016, PNAS. It would also be very interesting and simple to test microbiome composition of Arabidopsis mutants, such as the ref3. This would allow to assess how general is the described activity of 4-HCA.

In the publication by Becker et al. 2016, the mutation of the protein CCR (an enzyme that

catalyzes 4-HCA) in poplar tree caused the enrichment of 4-HCA in poplar tissues, and consequently increased the abundance of the bacterial genus *Pseudomonas*. We think that this discovery is consistent with our present finding that 4-HCA enriched *Pseudomonas* on rice leaves. We added the sentence: "A similar observation was obtained with poplar tree endophytes and down-regulation of cinnamoyl-CoA reductase; it caused the enrichment of 4-HCA in plant tissues and consequently increased the abundance of *Pseudomonas* (Becker et al. 2016)." in **line 502** in revised manuscript.

Related to the general applicability of our findings, we are already working on obtaining mutants from other plant species and intend to present the findings in a follow-up study due to the extent of the work required.

Reviewer #3 (Remarks to the Author):

Su and co-workers presented an investigation of the molecular basis regulating host-microbe interactions in the leaves of the staple crop rice. Combining an innovative genome-wide association study, using microbiota information as an "external" plant trait, with plant functional genomics, metabolomics and microbial inoculation they concluded that a precursor of plant lignin is required for the assembly of the endogenous microbiota which, in turn, fends off opportunistic pathogens in rice leaves. The manuscript reads well and, almost invariably, the conclusions represent an accurate interpretation of the data generated. What makes this story standing, in my vision, is the comprehensive and multidisciplinary approach: from gene identification, to validation and, perhaps most importantly, an effort to establish a clear causal relationships between plant genetic variation and configurations of the microbiota. For these reasons, I am confident this manuscript may represent a reference for future investigations of the plant microbiota in crop plants.

I attach a number of, relatively minor, questions and suggestions for the authors in the commented version of the manuscript.

1.Line102~103 bacterial taxonomies/abundances? ...in the rice genome ...significantly associated with...

We added the suggestion "we performed GWAS to link bacterial abundances with single nucleotide polymorphisms (SNPs) in the rice genome." in **line104** in revised manuscript.

We also implemented "Rice genetic variation was significantly associated with members of four predominant phyllosphere bacterial orders, Pseudomonadales, Burkholderiales, Enterobacterales, and Xanthomonadales" in **line 106** of the revised manuscript.

2.Line106~108 'To unravel a prevailing mechanism of how host genetics affect phyllosphere microbiome assembly, we implemented mutants and over- expression constructs of a candidate gene associated with Pseudomonadales and assessed the resulting microbiome shifts.' There is a flow leap here as it isn't immediately clear how the candidate genes relate to 4-HCA.

We modified the sentence in **line112** of the revised manuscript to read "Furthermore, we

analyzed rice metabolites in rice leaves that are regulated by the candidate gene.”

3.line109 “plays a key role in the assembly”. bit vague... what about 'is required for the assembly and homeostasis of at least a part of....'

We agree and have modified it to read “We discovered that the compound 4-hydroxycinnamic acid, a precursor in lignin biosynthesis, is required for the assembly and homeostasis of the rice phyllosphere microbiome.” in **line113** of the revised manuscript.

4.line128 “Bacterial communities obtained from INDICA and JAPONICA varieties formed two distinct clusters, as indicated by the unconstrained principal coordinate analysis (PCoA) of Bray-Curtis distances (Fig. 1A, $P < 0.001$, PERMANOVA with Adonis test).” What proportion? I mean was it 5% or 50% of the total variation? I'd add the R^2 value of the factor subspecies as this gives weight to the finding.

We added the R^2 value and revised the sentence as “Bacterial communities obtained from *INDICA* and *JAPONICA* varieties formed two distinct clusters, as indicated by an unconstrained principal coordinate analysis (PCoA) of Bray-Curtis distances (Fig. 1A, $P < 0.001$, $R^2 = 0.03$, PERMANOVA with Adonis test)” in **line 134** of the revised manuscript.

5.line 134 Sure? I mean Figure 1B depicts microbial evenness not richness. This statement needs to be verified (e.g., by computing a richness index). Marks from reviewer: “OK, now I found the relevant piece of information in the supplementary. Please add reference to Fig S2 here too.”

We revised the sentence as: “The phyllosphere microbiome of *INDICA* plants was more diverse than that of *JAPONICA* plants (Fig. 1B and Fig. S2), indicating that the former was colonized by more bacterial species.” in **line 139** of the revised manuscript.

6.line151 Fig. S4 very interesting graphical output!

Thank you, we used Fig. S4 to clearly show the significantly different relative abundances of bacterial genera in *INDICA* and *JAPONICA* varieties.

7.line212” while Burkholderiales-, Pseudomonadales- and Xanthomonadales-associated rice genes were enriched in the Phenylpropanoid biosynthesis pathway (dosa00940) (Fig. 2B, Table S9).” Key point

We agree, this was an important result that signified the key role of the phenylpropanoid biosynthesis pathway with respect to phyllosphere microbiome assembly.

8.line235-238 “Collectively, our analysis suggested that compounds that are part of the lignin biosynthesis or their precursors may be important for regulating microbiota assembly, and more importantly, that certain compounds may exert effects on specific bacterial taxa.” “Given the common occurrence and already known importance of members from the Pseudomonadales order in the plant phyllosphere” rephrase or identify an appropriate synonymous in those instances.

We revised the sentence as “Collectively, our analysis suggested that compounds that are part of lignin biosynthesis or their precursors may be participating in regulating microbiota assembly, and more importantly, that certain compounds may exert effects on specific bacterial taxa.” in **line241** of the revised manuscript.

We revised the sentence as “Given the common occurrence and already known significance of members from the Pseudomonadales order in the plant phyllosphere...” in **line 246** of the revised manuscript.

9.line242 “Pseudomonadales were not only highly abundant in the present study, but also accounted for the largest number of associated loci in the GWAS approach, indicating a robust connection to plant genetics.” ...or simply the mere consequence of the fact that Pseudomonadales is the most abundant order you identified....

Yes, the GWAS results are based on our metagenomic profiling of rice phyllosphere, in which Pseudomonadales accounted for the largest number of associated loci.

10.line249“Intriguingly, natural variation of SNPs in OsPAL02 was different between INDICA and JAPONICA rice (Fig. S7), suggesting a correlation between Pseudomonadales abundance and this gene.” Very interesting observation! Is it known how natural variation at this locus impacts on gene functionality/lignin biosynthesis?

This is an interesting point, but we don’t know if natural variation at this locus impacts lignin biosynthesis. We did not investigate this in detail in the present study, because our focus was on the microbiome. However, we did not observe obvious morphological difference between wild type and OE plants. Therefore, we assume there are no major consequences on lignin biosynthesis.

11.line262 “This result was consistent with initial analysis of the leaf microbiome, indicating that field observations were reproducible in the greenhouse.” Marks from reviewer: Key point. “Unconstrained PCoA of Bray-Curtis distances revealed that the WT, OsPAL02-OE and OsPAL02-KO phyllosphere microbiomes formed separated clusters, indicating that OsPAL02 affected their bacterial communities (Fig. 3B)” While I understand that unconstrained ordinations provide you with a global vision of community partitioning among genotypes this is the case where constrained ordinations enable you to weight the impact of a given factor, in this case OsPAL02 genotype. I'd suggest to include this analysis.

This is a good point. We added a constrained principal coordinate analysis (CPCoA). We revised the sentence as “Unconstrained PCoA (Fig. 3B, $P < 0.001$, $R^2 = 0.32$, PERMANOVA) and constrained principal coordinate analysis (CPCoA) (Fig. S8, 23.7% of total variance was explained by the plant genotype, $P < 0.001$, PERMANOVA) of Bray-Curtis distances revealed that the WT, *OsPAL02-OE* and *OsPAL02-KO* phyllosphere microbiomes formed separate clusters, indicating that *OsPAL02* affected their bacterial communities.” in **line 272** of the revised manuscript.

12.line275 “This result indicated that differences in OsPAL02 were responsible for the microbiome variations between INDICA and JAPONICA.” You identified hundreds of associations in the GWAS

and tested just one... I think you are running to fast in drawing conclusion. what about "were sufficient to mimic microbiome variations between indica and japonica" instead. The second question I have is what happens with the other associations? Do you think that those are a "consequence" of the PAL2 haplotype? I mean a population structure effect somehow dictate by PAL2? This is where it would be useful knowing the R^2 of the PERMANOVA or a constrained ordination as it will inform of the effect of PAL2 versus the other genetic background.

We revised the sentence as "This result indicated that differences in *OsPAL02* were sufficient to mimic microbiome variations between *INDICA* and *JAPONICA* plants." in **line 289** of the revised manuscript.

We also conducted Linkage Disequilibrium Analysis between *OsPAL02* and other GWAS loci and the resulting average R^2 value was 0.055. Among the 300+ GWAS loci, *OsPAL02* does not appear to be closely linked to other adjacent loci (shown in Figure 2 below) or to any of the other 300+ associated loci. Therefore, the associations of other loci are not consequences of the *OsPAL02* locus.

Figure 2. Linkage Disequilibrium Analysis between all GWAS loci. The red arrow indicates *OsPAL02* locus.

13.line279 "In the phenylpropanoid biosynthesis pathway" I'd insert a paragraph break here as you move from genetics to metabolites.

This is a good point. We also added the sentence "When *OsPAL02* variation between *JAPONICA* and *INDICA* was analyzed, the protein structures of haplotype 1 (mainly present in *INDICA*) and haplotype 5 (mainly present in *JAPONICA*) showed an amino acid exchange at site 134. This result

indicated that OsPAL02 catalytic activity may differ between *JAPONICA* and *INDICA* plants (Fig. S7)." after the sentence "In the phenylpropanoid biosynthesis pathway, ...".

14.line283 We therefore hypothesized that the products of OsPAL02 impose a selective...key point (and a very logic one)

Thanks. This was driven by the fact that significant changes in the relative abundance of Pseudomonadales were observed among the WT, KO and OE rice plants.

15.line291" The data was also subjected to unconstrained PCoA and the results showed that the first principal coordinate PCo1, which accounts for 53.89% of the total variance, separated the WT, OsPAL02-OE and OsPAL02-KO, indicating substantial differences in metabolites present in rice leaves (Fig. 4B)." same comment as before: you have a straightforward hypothesis you can test (and quantify) with a constrained ordination for the gene you modified.

We have included this analysis. We revised the sentences as "The data was also subjected to unconstrained PCoA and the results showed that the first principal coordinate PCo1, which accounts for 53.89% of the total variance, separated the WT, OsPAL02-OE and OsPAL02-KO plants, indicating substantial differences in metabolites present in rice leaves (Fig. 4B, $P < 0.001$, $R^2 = 0.60$, PERMANOVA). This was also observed with CPCoA (Fig. S10) where 46.1% of total variance was explained by the plant genotype ($P < 0.001$, PERMANOVA)." The revised sentences were included in **line 309** of the revised manuscript.

16.line300" Apart from the altered phyllosphere microbiomes, OsPAL02-KO developed severe disease symptoms compared with WT and OsPAL02-OE after leaf cutting (Fig. 3E)." you have to be more specific here (e.g., arrowheads) or provide alternative figures as I can't see such severe disease symptoms. Also, could it be something related to electrolytic leakage instead? I mean not a plant-microbe phenomenon. OK got my answer from the second point a few lines below.

Yes, we observed the severe disease symptoms in follow-up experiments and show it in the figures such as Fig. 6 and Fig. S13.

17.Line304 "absence of OsPAL02 caused the dysbiosis in the rice phyllosphere microbiome and consequently led to the vulnerability of plants to pathogens." ...on detached leaves though, not whole plants.

We revised the sentence to read ".....absence of OsPAL02 caused the dysbiosis in the rice phyllosphere microbiome and consequently led to the vulnerability of plants to pathogens on leaves."

18.line322, 625 "4-HCA plays a key role in maintaining plant health."...but you can produce seeds of OsPAL02-KO under non-sterile conditions, correct? This means that this pathogen protection effect does not impact on plant fitness or? see comment above about pathogen protection of OsPAL02

We cultivated the KO and OE lines in a propagation site for transgenic plants where pesticides

were applied in order to allow us to produce seeds. However, we observed that the seed setting rate was significantly reduced in KO lines (shown below in Figure 3). This result does not impact the main results, so we did not include it in the manuscript.

Figure 3. Rice panicles of WT, KO, and OE plants (A) and seed setting rate of the same plants (B).

19.Line336 “of the orders (relative abundance > 0.1%) detected with amplicon analyses (Fig. S12, Table S14).” Not clear what do you mean with that. Is this based on the identity of the 16S rRNA gene amplicon sequencing vs culture collection or just taxonomic affiliation?

We revised the sentence in line 370 as “They covered 62.5% of the orders (relative abundance > 0.1%) detected with amplicon sequencing analyses (Fig. S16, Table S16)”.

20.Line349, “As expected, the inoculation of TJ1 caused severe disease symptoms on OsPAL02-KO, meanwhile, OsPAL02-OE were less affected in terms of disease occurrence and fresh weight, compared to WT and OsPAL02-KO” Have you attempted at re-isolating TJ1 from inoculated plants? I mean is there a correlation between disease symptoms and microbial colonization in the different genotypes? Line 354 “was confirmed to be a rice pathogen” If you can re-isolate them...At the moment the only thing you can confirm is that inoculation of germ-free leaves with TJ1 induces symptoms in a (host) genotype-specific manner.

Furthermore, we re-isolated bacteria from TJ1-inoculated WT, OsPAL02-KO and OsPAL02-OE plants. Sequencing of their 16S rRNA genes confirmed 100% sequence similarity with TJ1 (Fig. S17). Additional isolation experiments showed that TJ1 abundance in OsPAL02-OE plants was significantly lower than in the OsPAL02-KO mutant and WT plants (Fig. S17). This provided further evidence that TJ1-induced disease resulted in the observed reduced average plant weight. Overall, the isolate TJ1 was confirmed to be a rice pathogen, and it was demonstrated to cause lighter disease symptoms in OsPAL02-OE than that observed with OsPAL02-KO and WT plants. The description of the results was added in line 388 of the revised manuscript.

21.line368 “faster in the presence of 2 mM 4-HCA, whereas all Xanthomonadales and Burkholderiales strains were significantly inhibited (Fig. 5A).” Interesting!

Indeed, we also found it interesting that Pseudomonadales strains outperformed other tested strains in the presence of 2 mM 4-HCA.

22.line395" Under gnotobiotic conditions, WT, OsPAL02-KO and OsPAL02-OE plants showed no significant differences in fresh weight when treated with either 10 mM MgCl₂ (control, Fig. 6A, B) or the SynCom suspension (Fig. S14). This result confirmed that the SynCom did not have a negative impact on rice plants" Key point

This result was important to prove that the SynCom had protective effects against TJ1.

23.Line407. "Evidently, an interplay between OsPAL02 and Pseudomonadales members was required for the SynCom-associated plant health maintenance" However, how do you explain that this mechanism hasn't (yet) been "hijacked" by opportunistic rice pathogens within the order (Pseudomonadales)? It would be very interesting checking whether a SynCom with potentially pathogenic Pseudomonadales may profit the same way from the production of 4-HCA.

This is a good point. We plan to investigate whether/how opportunistic rice pathogens within the order Pseudomonadales affect plant health maintenance and intend to present the findings in a follow-up study due to the required extent.

24.Line845 "All other raw data for all figures and tables are available from the corresponding authors upon reasonable request." I think this is unreasonable... I strongly recommend to have scripts on github and intermediate files, when not included among the material of the manuscript, on zenodo or similar, open access, platforms.

We added a Code Availability section in **line 923** of the revised manuscript:

Scripts employed in the computational analyses are available at:
<https://github.com/kanghouxiang105/micro-GWAS>

In addition, I'd add a provision for the transformed lines and the rice bacterial collection as readers may be interested in acquiring them for follow-up investigations.

We added the sentence in **line 920** of the revised manuscript "All isolated bacterial strains and transformed rice lines were deposited in the State Key Laboratory of Hybrid Rice and the Institute of Plant Protection of Hunan Academy of Agricultural Sciences (Changsha, China)."

Reviewers' Comments:

Reviewer #1:

Remarks to the Author:

The authors have adequately addressed my previous comments. I have no further comments.

Reviewer #2:

Remarks to the Author:

I am happy with the way the authors addressed my comments in the revised manuscript.

Reviewer #3:

Remarks to the Author:

In this revised version, Su and co-workers adequately addressed all my criticisms: I have no further comments for them.

I found the first version of the manuscript quite exciting to read, I do believe that the revisions strengthened the robustness of the findings, likely making this manuscript a reference framework for future investigations into the genetics of host-microbiota interactions in crop plants.

Response to reviewers

Reviewer #1 (Remarks to the Author):

The authors have adequately addressed my previous comments. I have not further comments.

Reviewer #2 (Remarks to the Author):

I am happy with the way the authors addressed my comments in the revised manuscript.

Reviewer #3 (Remarks to the Author):

In this revised version, Su and co-workers adequately addressed all my criticisms: I have no further comments for them.

I found the first version of the manuscript quite exciting to read, I do believe that the revisions strengthened the robustness of the findings, likely making this manuscript a reference framework for future investigations into the genetics of host-microbiota interactions in crop plants.

Response to reviewer's comments

We would like to express our sincere gratitude to all reviewers for their invaluable feedback, which we firmly believe significantly contributed to improve the final version of the paper.